# Traversal Verification for Speculative Tree Decoding

Yepeng Weng[1]*    Qiao Hu[2]*†    Xujie Chen[1]    Li Liu[1]
Dianwen Mei[1]    Huishi Qiu[1]    Jiang Tian[1]    Zhongchao Shi[1]
[1]Lenovo AI Technology Center, Lenovo
[2] National Center for Mathematics and Interdisciplinary Sciences (NCMIS), AMSS, CAS

## Abstract

Speculative decoding is a promising approach for accelerating large language models. The primary idea is to use a lightweight draft model to speculate the output of the target model for multiple subsequent timesteps, and then verify them in parallel to determine whether the drafted tokens should be accepted or rejected. To enhance acceptance rates, existing frameworks typically construct token trees containing multiple candidates in each timestep. However, their reliance on token-level verification mechanisms introduces two critical limitations: First, the probability distribution of a sequence differs from that of individual tokens, leading to suboptimal acceptance length. Second, current verification schemes begin from the root node and proceed layer by layer in a top-down manner. Once a parent node is rejected, all its child nodes should be discarded, resulting in inefficient utilization of speculative candidates. This paper introduces *Traversal Verification*, a novel speculative decoding algorithm that fundamentally rethinks the verification paradigm through leaf-to-root traversal. Our approach considers the acceptance of the entire token sequence from the current node to the root, and preserves potentially valid subsequences that would be prematurely discarded by existing methods. We theoretically prove that the probability distribution obtained through Traversal Verification is identical to that of the target model, guaranteeing *lossless* inference while achieving substantial acceleration gains. Experimental results on various models and multiple tasks demonstrate that our method consistently improves acceptance length and throughput over token-level verification.

## 1   Introduction

Large Language Models (LLMs) have been widely adopted due to their exceptional performance across various natural language processing tasks [10, 25, 34] . However, the massive parameters and the autoregressive generation scheme of transformer decoder-only [30] LLMs limit the generation speed. Speculative decoding [19, 3] is an lossless acceleration technique which employs a lightweight model (draft model) with fewer parameters to speculate the output tokens of the original LLM (target model) for several future timesteps, then feed the drafted tokens into the target model in parallel. After getting the probability distribution of the target model, speculative decoding determines the acceptance or rejection of each token based on their probabilities in both target and draft models. If a token is rejected, a new token will be resampled and all subsequent tokens should be discarded.

To further improve acceleration performance, existing methods [23, 4, 21, 14, 35] generate multiple candidates at each drafting timestep, forming a tree of drafted tokens. However, these methods generally inherit the token-level verification mechanism from vanilla speculative decoding to tree scenarios, resulting in suboptimal acceptance lengths in tree decoding. To be more specific, firstly,

---

*Equal contribution. Contact: wengyp1@lenovo.com, huqiao2020@amss.ac.cn
†Corresponding author.

the probability distribution of a token sequence differs from that of an individual token. Vanilla speculative decoding determines acceptance based on per-token probabilities, which sacrifices global optimality for sequence-level acceptance. Secondly, existing tree decoding methods start verification from the root node of the tree, and proceed layer by layer in a top-down manner. Once a parent node is rejected, all its child nodes will be discarded accordingly, resulting in the wasting of drafted tokens.

To address these issues, we propose a novel speculative decoding method named *Traversal Verification*. Unlike existing methods, Traversal Verification starts from the leaf node and generally operates in a bottom-up manner. If the node is accepted, the entire sequence from the current node to the root is accepted. If rejected, the algorithm proceeds to verify the sibling nodes (or the deepest child nodes of its siblings if they exist). If all siblings are rejected, it backtracks to the parent node. This process repeats until either a node is accepted or all nodes in the tree are rejected.

Through Traversal Verification, we effectively resolve the limitations of existing methods. First, we consider sequence-level probabilities instead of individual token probabilities and improve the acceptance lengths. Second, in Traversal Verification, a parent node will be verified only after all its child nodes have been rejected, which minimizes the wasting of drafted candidates.

We conducted experiments on Llama3 [10] series and Llama2 [29] using various tree structures. The experiments were performed on the Spec-Bench dataset [32], which encompasses six different tasks: multi-turn conversation, translation, summarization, question answering, mathematical reasoning, and retrieval-augmented generation. Experimental results demonstrate that Traversal Verification consistently outperforms existing decoding methods by 2.2%-5.7% in average acceptance length across diverse tasks with different tree architectures. Additionally, Traversal Verification could potentially achieve greater improvements for deeper and larger decoding trees.

We highlight the advantages of Traversal Verification as follows:

1. **Full utilization of drafted tokens.** Traversal Verification enhances acceptance length and improves the utilization of drafted tokens by considering sequence-level probability distributions and systematically traversing nodes in the token tree. To our knowledge, it is the *first* verification algorithm that makes use of the whole token tree.

2. **Reliable generation quality.** We theoretically prove that Traversal Verification is a *lossless* verification algorithm, that is, the output distribution is identical to that of the target model. This serves as a powerful guarantee of generation quality.

3. **Pronounced improvement.** Experiments across various tree structures and datasets shows that Traversal Verification outperforms token-level verification. We also rigorously prove that Traversal Verification is *theoretically optimal* in the case of a single chain.

4. **Minimal implementation modification.** Traversal Verification serves as a *plug-and-play* replacement of existing verification methods. There is no need to change other parts of existing speculative decoding pipelines.

## 2 Preliminaries

### 2.1 Speculative Decoding

Speculative decoding, also known as speculative sampling [19, 3], is a lossless LLM acceleration algorithm. In speculative decoding, a draft model first generates a chain of $\gamma$ new tokens (*i.e.,* one token per timestep for the next $\gamma$ timesteps), then the drafted tokens are fed into the target model in parallel to get the target distribution.

We denote the drafted token chain by $\alpha^\gamma = (\alpha_0, \alpha_1, \ldots, \alpha_\gamma)$, where $\alpha_0$ represents the prefix and $\alpha_{>0} := (\alpha_1, \ldots, \alpha_\gamma)$ denotes the $\gamma$ new tokens generated by the draft model. After obtaining the target distribution $\mathcal{M}_b$, the drafted tokens will be verified from timestep 1 to $\gamma$ following Algorithm 1.

---

**Algorithm 1** Single-token verification

**Input:** Prefix $X_0$; draft token $X$; draft distribution $\mathcal{M}_s(\cdot|X_0)$; target distributions $\mathcal{M}_b(\cdot|X_0)$ and $\mathcal{M}_b(\cdot|X_0, X)$.
1: Sample $\eta \sim U(0,1)$.
2: **if** $\eta < \frac{\mathcal{M}_b(X|X_0)}{\mathcal{M}_s(X|X_0)}$ **then**
3:     Sample $Y$ from $\mathcal{M}_b(\cdot|X_0, X)$.
4:     **Return:** $X, Y$.
5: **else**
6:     Sample $Y$ from $\text{norm}([\mathcal{M}_b - \mathcal{M}_s]_+)$.
7:     **Return:** $Y$.
8: **end if**

---

|         | $\mathcal{M}_s$ | $a$ | $b$ | $c$ |
|---------|-----|-----|-----|-----|
| $\mathcal{M}_b$ |     | 0.6 | 0.3 | 0.1 |
| $a$     | 0.3 | 0.3 | 0   | 0   |
| $b$     | 0.4 | 0.1 | 0.3 | 0   |
| $c$     | 0.3 | 0.2 | 0   | 0.1 |

Table 1: An example of single-token verification

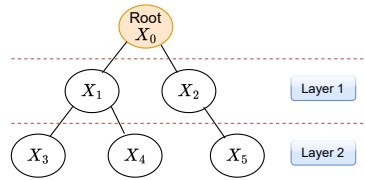

Figure 1: An example token tree

If a token is accepted, the verification proceeds to the next timestep. Once a token is rejected, all subsequent tokens in the chain are discarded, and a new token will be resampled at the rejection position based on the residual probability distribution. If all $\gamma$ tokens are accepted, an additional token is sampled from the target distribution at the timestep $\gamma + 1$. The output of a drafting-verification cycle thus consists of all accepted tokens plus the resampled or newly sampled token at the final step.

To illustrate the acceptance mechanism intuitively, consider a simplified example with a vocabulary of three tokens: $[a, b, c]$. Let the target model's probability distribution be $\mathcal{M}_b = [0.3, 0.4, 0.3]$, and the draft model's distribution be $\mathcal{M}_s = [0.6, 0.3, 0.1]$. All possible cases are summarized in Table 1.

According to Algorithm 1, if token $b$ is sampled, it will be accepted directly because $\mathcal{M}_b(b) > \mathcal{M}_s(b)$. Similarly, $c$ will be accepted if sampled. If $a$ is sampled, the acceptance probability is $\frac{\mathcal{M}_b(a)}{\mathcal{M}_s(a)} = 0.5$. Thus, the probability of generating token $a$ is $\mathbb{P}(\text{sample } a) \times \mathbb{P}(\text{accept } a) = 0.3$, which is equal to $\mathcal{M}_b(a)$. These cases correspond to the diagonal entries in Table 1, highlighted in green.

Besides being accepted, token $a$ also faces a rejection probability of 0.5. Upon rejection, a new token is resampled from the residual probability distribution $\text{norm}([\mathcal{M}_b - \mathcal{M}_s]_+)$. Specifically, we subtract $\mathcal{M}_s$ from $\mathcal{M}_b$ and set the negative values to zero (yielding $[0, 0.1, 0.2]$ in this example), and then normalize the residual probabilities. Therefore, the final probabilities for $b$ and $c$ consist of two parts: 1) direct acceptance after sampling from $\mathcal{M}_s$ and 2) resampling after rejection of $a$, indicated in cyan in Table 1. By this means, the final distribution is kept identical to $\mathcal{M}_b$.

## 2.2 Recursive Rejection Sampling

Recursive Rejection Sampling (RRS) samples multiple candidates at each timestep and recursively verifies them, as described in Algorithm 2. Recent works [4, 21, 14, 35] further refine RRS into RRS without replacement (RRSw), where the probability of a rejected token in $\mathcal{M}_s$ is set to zero, and then normalize $\mathcal{M}_s$ of the remaining candidates. RRSw prevents repeated sampling and rejection of the same token, especially for low-temperature situations, thereby improving overall acceptance rates.

We illustrate RRSw using the same example in Table 1. Suppose that token $a$ is sampled and rejected. The residual distribution becomes $\mathcal{M}'_b = \text{norm}([\mathcal{M}_b - \mathcal{M}_s]_+) = [0, 1/3, 2/3]$, while the new draft distribution $\mathcal{M}'_s = \text{norm}(\mathcal{M}_s(a) = 0) = [0, 3/4, 1/4]$. Then we sample a new token from $\mathcal{M}'_s$ and repeat the speculative decoding scheme: If token $b$ is sampled, it is accepted with probability $\frac{\mathcal{M}'_b(b)}{\mathcal{M}'_s(b)} = 4/9$. If $c$ is sampled, it is always accepted since $\mathcal{M}'_b(c) > \mathcal{M}'_s(c)$. For scenarios with more candidates, this process iterates until all candidates are verified.

Combining chain-based speculative decoding with multi-candidate per timestep yields tree decoding. In the current framework, candidate tokens are verified layer by layer from shallow to deep: if a node is rejected, we continue to verify its siblings; the current node itself and all its children are discarded. If a node is accepted, the verification proceeds to its child nodes in the deeper layer.

---

**Algorithm 2** Recursive Rejection Sampling

**Input:** Prefix $X_0$; draft distribution $\mathcal{M}_s(\cdot|X_0)$; $k$ drafted candidates $\{X_i\}_{i=1}^k$ from $\mathcal{M}_s(\cdot|X_0)$; target distributions $\mathcal{M}_b(\cdot|X_0)$ and $\mathcal{M}_b(\cdot|X_0, X_i), \forall 1 \leqslant i \leqslant k$.

1: Initialize residual $\mathcal{M}'_b$ with $\mathcal{M}_b(\cdot|X_0)$ and draft $\mathcal{M}'_s$ with $\mathcal{M}_s(\cdot|X_0)$.
2: **for** i=1,...,k **do**
3:     Sample $\eta \sim U(0, 1)$.
4:     **if** $\eta < \frac{\mathcal{M}'_b(X_i)}{\mathcal{M}'_s(X_i)}$ **then**
5:         Sample $Y$ from $\mathcal{M}_b(\cdot|X_0, X_i)$.
6:         **Return:** $X_i, Y$.
7:     **else**
8:         $\mathcal{M}'_b \leftarrow \text{norm}([\mathcal{M}'_b - \mathcal{M}'_s]_+)$.
9:         $\mathcal{M}'_s \leftarrow \text{norm}(\mathcal{M}'_s(X_i) = 0)$ (if without replacement)
10:    **end if**
11: **end for**
12: Sample $Y$ from $\mathcal{M}'_b(\cdot|X_0)$.
13: **Return:** $Y$.

---

We demonstrate the token-level verification order using a simplified two-layer decoding tree, as shown in Figure 1. In this tree, node $X_1$ is verified first. If accepted, we proceed to its children ($X_3$ and $X_4$) and verify them sequentially. If $X_1$ is rejected, we discard $X_1$, $X_3$, $X_4$, and go to $X_2$. If $X_2$ is accepted, we continue to verify $X_5$, otherwise, since all the sampled tokens are rejected, we will resample a new token from the residual probability distribution of Layer 1.

## 3 Method

In this section, we first introduce Traversal Verification. Subsequently, we illustrate its distinctions from token-level tree decoding (vanilla speculative decoding with RRSw) through an intuitive example (see Figure 2). In the last part of this section, we discuss the theoretical guarantees, such as the losslessness of Traversal Verification, and its optimality in single chain scenarios.

### 3.1 Traversal Verification

We present Traversal Verification in Algorithm 3.

---

**Algorithm 3** Traversal Verification

---

**Input:** Prefix $X_0$ as the root; a valid sampling tree $T$ on draft distribution $\mathcal{M}_s$; for all chains $\forall \alpha = (X_0, \ldots, X_{\gamma_\alpha}) \subset T$, draft distributions $\forall i < \gamma_\alpha$, $\mathcal{M}_s(\cdot|X^i)$ and target distributions $\forall i \leqslant \gamma_\alpha$, $\mathcal{M}_b(\cdot|X^i)$.

1: **Initialize:** For all chains $\forall \alpha = (X_0, \ldots, X_{\gamma_\alpha}) \subset T$, let $p_\alpha^{ini}(X_0) = 1$ and then recursively set the acceptance rates for all nodes of $\alpha$,

$$p_\alpha^{ini}(X_i) = \min\left\{p_\alpha^{ini}(X_{i-1})\frac{\mathcal{M}_b(X_i|X^{i-1})}{\mathcal{M}_s(X_i|X^{i-1})}, 1\right\}, \quad 1 \leqslant i \leqslant \gamma_\alpha.$$

2: Set $p_\alpha(X_i) = p_\alpha^{ini}(X_i), \forall X_i \in \alpha, \forall \alpha \subset T$, and the acceptance length $\tau = 0$
3: **while** $T \neq \emptyset$ **do**
4:     Select $\alpha = (X_0, \ldots, X_{\gamma_\alpha}) \subset T$ from root to the first leaf node, with $\gamma_\alpha$ being its depth.
5:     Sample $\eta \sim U(0, 1)$.
6:     **if** $\eta < p_\alpha(X_{\gamma_\alpha})$ **then**
7:         $\tau = \gamma_\alpha$ and $X^\tau = (X_0, \ldots, X_{\gamma_\alpha})$.
8:         **break.**
9:     **else**
10:         Delete the last node of $\alpha$ from the tree $T$, that is $T \leftarrow T - \{X_{\gamma_\alpha}\}$.
11:         Set the residual and draft distributions by (1) and (2), *i.e.,*

$$\mathcal{M}_b'(x|X^{\gamma_\alpha-1}) = \text{norm}([p_\alpha(X_{\gamma_\alpha-1})\mathcal{M}_b(x|X^{\gamma_\alpha-1}) - \mathcal{M}_s(x|X^{\gamma_\alpha-1})]_+),$$

$$\mathcal{M}_s'(x|X^{\gamma_\alpha-1}) = \text{norm}(\mathcal{M}_s'(X_{\gamma_\alpha}|X^{\gamma_\alpha-1}) = 0).$$

12:         Set $p_\alpha'(X_{\gamma_\alpha-1})$ as (3) and then modify

$$p_\alpha(X_{\gamma_\alpha-1}) \leftarrow p_\alpha'(X_{\gamma_\alpha-1}),$$

$$\mathcal{M}_b(x|X^{\gamma_\alpha-1}) \leftarrow \mathcal{M}_b'(x|X^{\gamma_\alpha-1}), \quad \mathcal{M}_s(x|X^{\gamma_\alpha-1}) \leftarrow \mathcal{M}_s'(x|X^{\gamma_\alpha-1}), \quad \forall x \in \mathcal{X}$$

13:         Update the acceptance rates for remaining chains $\beta = (x_0, \ldots, x_{\gamma_\beta}) \subset T$ with the starting nodes $x^{\gamma_\alpha-1} = X^{\gamma_\alpha-1}$,

$$p_\beta(x_i) = \min\left\{p_\beta(x_{i-1})\frac{\mathcal{M}_b(x_i|x^{i-1})}{\mathcal{M}_s(x_i|x^{i-1})}, 1\right\}, \quad \gamma_\alpha \leqslant i \leqslant \gamma_\beta.$$

14:     **end if**
15: **end while**
16: Sample $Y$ from $\mathcal{M}_b(\cdot|X^\tau)$.
17: **Return:** $X^\tau, Y$.

---

Residual distribution in Algorithm 3 (Line 11): $\forall x \in \mathcal{X}$,

$$\mathcal{M}_b'(x|X^{\gamma_\alpha-1}) = \frac{[p_\alpha(X_{\gamma_\alpha-1}) \cdot \mathcal{M}_b(x|X^{\gamma_\alpha-1}) - \mathcal{M}_s(x|X^{\gamma_\alpha-1})]_+}{\sum_x [p_\alpha(X_{\gamma_\alpha-1}) \cdot \mathcal{M}_b(x|X^{\gamma_\alpha-1}) - \mathcal{M}_s(x|X^{\gamma_\alpha-1})]_+}. \qquad (1)$$

Modified draft distribution in Algorithm 3 (Line 11): $\forall x \in \mathcal{X}$,

$$\mathcal{M}_s'(X_{\gamma_\alpha}|X^{\gamma_\alpha-1}) = 0 \text{ and } \mathcal{M}_s'(x|X^{\gamma_\alpha-1}) \leftarrow \frac{\mathcal{M}_s(x|X^{\gamma_\alpha-1})}{1 - \mathcal{M}_s(X_{\gamma_\alpha}|X^{\gamma_\alpha-1})} \text{ if } x \neq X_{\gamma_\alpha}. \quad (2)$$

Acceptance probability in Algorithm 3 (Line 12):

$$p_\alpha'(X_{\gamma_\alpha-1}) = \frac{\sum_x [p_\alpha(X_{\gamma_\alpha-1}) \cdot \mathcal{M}_b(x|X^{\gamma_\alpha-1}) - \mathcal{M}_s(x|X^{\gamma_\alpha-1})]_+}{\sum_x [p_\alpha(X_{\gamma_\alpha-1}) \cdot \mathcal{M}_b(x|X^{\gamma_\alpha-1}) - \mathcal{M}_s(x|X^{\gamma_\alpha-1})]_+ + 1 - p_\alpha(X_{\gamma_\alpha-1})} \qquad (3)$$

Traversal Verification exhibits two key distinctions from token-level tree decoding:

1. **Bottom-up verification**. Traversal Verification generally operates in a bottom-up manner, starting verification from leaf nodes (*i.e.,* deeper layers) and progressing toward the root, while token-level tree decoding follows a top-down approach, verifying nodes layer by layer from shallow to deep. Details about traversal order are provided in Appendix E.

2. **Sequence-level acceptance**. Traversal Verification incorporates the joint probability distribution of the token sequence, rather than relying solely on per-token probabilities. The acceptance rate at each node represents the sequence-level acceptance rate from the current node to the root. Thus, once a token is accepted, the entire sequence from the current node to root is accepted.

## 3.2 An Intuitive Example of Traversal Verification

We now demonstrate Traversal Verification using the same illustrative case as introduced in Section 2.2. Following Algorithm 3, for the tree structure in Figure 1, the traversal order is $X_3 \rightarrow X_4 \rightarrow X_1 \rightarrow X_5 \rightarrow X_2$. Consider a tree with nodes sampled as $[X_1, X_2, X_3, X_4, X_5] = [a, c, b, c, a]$ as an intuitive example. We present the detailed process of Traversal Verification in Figure 2.

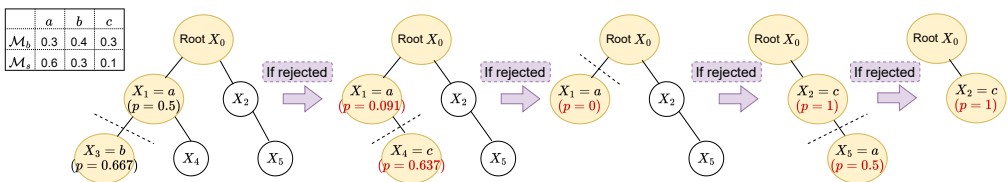

Figure 2: The traversal order of verifying a sampling tree.

We define $r(X_i) = \frac{\mathcal{M}_b(X_i)}{\mathcal{M}_s(X_i)}$ for simplification. For the first chain $X_1 X_3$, the acceptance rate of Traversal Verification is

$$\mathbb{P}_{\text{traversal}}(\text{accept } X_1 X_3) = \min\left(\min(r(X_1), 1) \cdot r(X_3), 1\right) = \min(0.5 \cdot 0.4/0.3, 1) \approx 0.667,$$

However, in token-level verification, the acceptance probability is only

$$\mathbb{P}_{\text{token-level}}(\text{accept } X_1 X_3) = \min(r(X_1), 1) \cdot \min(r(X_3), 1) = 0.5.$$

When $X_1 X_3$ is rejected, we delete the last node $X_3$ and then the first chain becomes $X_1 X_4$. According to Line 11-13 in Algorithm 3, since $[p(X_1)\mathcal{M}_b(a) - \mathcal{M}_s(a)]_+ = 0$ and $[p(X_1)\mathcal{M}_b(c) - \mathcal{M}_s(c)]_+ = 0.05$, the new $p'(X_1)$ and the acceptance rate of chain $X_1 X_4$ are updated as

$$p'(X_1) = \frac{0.05}{0.05 + 1 - 0.5} \approx 0.091,$$

and

$$\mathbb{P}(\text{accept } X_1 X_4) = \min\left(p'(X_1) \cdot \frac{\mathcal{M}_b'(X_4)}{\mathcal{M}_s'(X_4)}, 1\right) \approx 0.637.$$

If $X_4$ is rejected, the residual acceptance probability of $X_1$, namely $p'(X_1)$, is reduced to zero, indicating that it cannot be accepted any more and should be removed immediately.

After node $X_1$ is discarded, the draft and target distributions of token-level verification and Traversal Verification in Layer 1 return to the same starting line once again. Then for the single chain $X_2 X_5$, Traversal Verification still expects longer acceptance than token-level verification (see Theorem 3.4).

### 3.3 Theoretical Guarantees

In this section, we formally establish the theoretical guarantees of Traversal Verification. Specifically, we prove that the following statements hold for Traversal Verification:

1. Traversal Verification is a *valid* (*i.e., lossless*) tree verification algorithm, which means the probability distribution of output sequences is identical to that of the target model.

2. In the special case where the sampling tree is a single chain, Traversal Verification achieves the optimal upper-bound of expectation of acceptance length.

We first formally define the decoding tree under autoregressive generation as follows:

**Definition 3.1** (Decoding tree under autoregressive generation). *Let $\mathcal{M}_s$ be a given distribution and $T$ be a decoding tree rooted at $X_0$ under autoregressive generation. For all chains $v = (X_0, \ldots, X_{\gamma_v}) \subset T$ where $\gamma_v$ denotes the depth of chain $v$, if all child nodes of $v$ are generated according to the conditional distribution $\mathcal{M}_s(\cdot|v)$ (with or without replacement), then the sampling tree $T$ is termed a decoding tree based on $\mathcal{M}_s$ under autoregressive generation.*

*For brevity, we hereafter refer to a tree satisfying the above definition as a* decoding tree.

Given an decoding tree $T$, we prove that Traversal Verification serves as a valid tree verification algorithm. A valid tree verification algorithm is defined as follows:

**Definition 3.2** (Valid tree verification algorithm). *Let $T$ be a decoding tree defined as Definition 3.1. For all chains $v = (X_0, \ldots, X_{\gamma_v}) \subset T$ with depth $\gamma_v$, a tree verification algorithm $\mathcal{A}_{\text{ver}}$ takes the tree $T$, draft model distributions $\mathcal{M}_s(\cdot|X^i)$, $\forall i < \gamma_v$ and target model distributions $\mathcal{M}_b(\cdot|X^i)$, $\forall i \leqslant \gamma_v$ as inputs, and outputs an accept chain $X^\tau = (X_0, \ldots, X_\tau) \subset T$ where $\tau \leqslant \max_{v \subset T} \gamma_v$ and an additional token $Y$.*

*The tree verification algorithm $\mathcal{A}_{\text{ver}}$ is called valid if its output distribution satisfies*

$$(X^\tau, Y) = \mathcal{A}_{\text{ver}}(T, \mathcal{M}_s, \mathcal{M}_b) \sim \mathcal{M}_b(\cdot|X_0), \tag{4}$$

*where $\mathcal{M}_b(X_0|X_0) = 1$.*

*Additionally, the tree verification $\mathcal{A}_{\text{ver}}$ is also called a valid chain verification algorithm if $T$ is a single chain and $\mathcal{A}_{\text{ver}}$ satisfies* (4).

For example, SpecInfer [23, Theorem 4.2] is a valid tree verification algorithm. In the case where the sampling tree degenerates into a single chain, both the vanilla token verification [19, Appendix.A] and Block Verification [27, Theorem 1] are valid chain verifications.

We now claim that Traversal Verification is a valid tree verification algorithm and is an optimal valid chain verification algorithm with $T$ being a single chain.

**Theorem 3.3** (Losslessness of Traversal Verification). *Traversal Verification (Algorithm 3) is a valid tree verification algorithm.*

**Theorem 3.4.** *When the sampling tree reduces to one single chain, for any valid chain verification algorithm VERIFY in Definition 3.2, let $N_{\text{traversal}}$, $N_{\text{block}}$ and $N_{\text{verify}}$ be the number of accepted tokens in Traversal Verification, Block Verification [27] and VERIFY, respectively, then for any given distributions $\mathcal{M}_s, \mathcal{M}_b$ and draft chain $T$, we have*

$$\mathbb{E}[N_{\text{traversal}}] = \mathbb{E}[N_{\text{block}}] \geqslant \mathbb{E}[N_{\text{verify}}],$$

*where $\mathbb{E}$ denotes the expectation taken over the randomness of draft chain $T$ and internal random variables utilized within the verification algorithms.*

**Discussions on theoretical foundations and design motivation of Traversal Verification.** The core idea of proving the losslessness (Theorem 3.3) of Traversal Verification lies in exploiting its

self-similarity. The self-similarity of Traversal Verification implies that, for any parent node $A$ in the given sampling tree $T$, before determining the acceptance of $A$, all its descendant nodes have already been processed through the same traversal mechanism. In other words, every local subtree within the sampling tree $T$ essentially operates as a scaled-down instance of the Traversal Verification mechanism. Consequently, we can employ mathematical induction on the number of descendant nodes to establish the critical Lemma A.2, from which Theorem 3.3 (the lossless theorem) directly follows as a corollary.

For the single-chain optimality of Traversal Verification (Theorem 3.4), the key proof idea is to ensure that Traversal Verification achieves the highest possible acceptance probability at each node, aligning with Block Verification. Assume that the acceptance rate for a parent node $A$ is $P(A)$. As a bottom-up verification framework, the target probability distribution for child nodes of $A$ should be $P(A)\mathcal{M}_b$. By introducing a pseudo-child node with target probability $(1 - P(A))$, we can apply RRSw to transport the draft distribution $\mathcal{M}_s$ to the target distribution $P(A)\mathcal{M}_b$ combining with $(1 - P(A))$. We refer to the above process as the *sequence-level RRSw method*. Comprehensive details are provided in Appendix F. This motivation directly leads to the formulations (1)–(3) of Traversal Verification. Since Block Verification is an optimal valid chain verification algorithm [27, Theorem 2], Traversal Verification inherits this optimality in the single-chain case (see Theorem 3.4).

## 4   Experiments

### 4.1   Experimental Setup

**Target LLMs and draft model.**   We mainly conduct experiments on the Llama3 [10] series, using Llama3.2-1B-Instruct as the draft model and Llama3.1-8B-Instruct as the target model. We also include Llama-68M [23] with Llama2-7b [29] as the draft and target model, which is widely adopted in existing speculative decoding researches [4, 12, 13, 26].

**Tasks.**   We perform experiments on the Spec-Bench dataset [32], which includes 80 instances from each of six distinct domains: multi-turn conversation (MT-Bench [36]), translation (WMT14 DE-EN [1]), summarization (CNN/Daily Mail [24]), question answering (Natural Questions [18]), Mathematical reasoning (GSM8K [5]), retrieval-augmented generation (DPR [16]).

**Metrics.**   We evaluate the performance of our method using two metrics: acceptance length and token generation speed. Acceptance length is the number of tokens generated per drafting-verification cycle, which reflects the theoretical performance of the verification method. We also include the actual throughput for a comprehensive comparison. It is worth noting that there may be slight variations in acceptance length according to differences in statistical methods, and we provide detailed discussions and additional experimental results on this issue in Appendix D.

**Implementation.**   For token-level tree verification, we adopt the RRSw implementation in EAGLE [21] from Spec-Bench [32] open source repository. All experiments are conducted on a single NVIDIA RTX A6000 GPU with PyTorch backend. Due to inherent randomness in sampling, we conduct three independent runs for each case and report the average as the result.

**Measurement of Generation Quality.**   Traversal Verification is theoretically a lossless speculative decoding technique, which suggests that evaluating its generation quality should not be mandatory. However, recognizing that some readers may seek assurance regarding this guarantee, we present the measurements of generation quality as a supporting reference for losslessness. Please consult Appendix C for the detailed experimental findings.

### 4.2   Overall Effectiveness

We present the acceptance lengths and throughput of two combinations of draft and target model, namely Llama3.2-1B-Instruct with Llama3.1-8B-Instruct and Llama-68M with Llama2-7B in Table 2 and Table 3. For chain and binary tree, we set the depth at 5, which is equal to the maximum depth of EAGLE sparse tree. Tok.V denotes token-level verification and Tra.V denotes Traversal Verification. The acceptance lengths are rounded to 2 decimal places, and we also provide the standard errors. $\Delta$ denotes the relative improvement of Traversal Verification over token-level verification. The baseline

Table 2: Acceptance length and throughput on Llama3.2-1B-Instruct with Llama3.1-8B-Instruct.

| | Llama3.2-1B-Instruct (draft) & Llama3.1-8B-Instruct (target) Temperature=1 | | | | | | | | |
| --- | --- | --- | --- | --- | --- | --- | --- | --- | --- |
| | Chain | | | Binary Tree | | | EAGLE Sparse Tree | | |
| Tasks | Tok.V | Tra.V | $\Delta$ | Tok.V | Tra.V | $\Delta$ | Tok.V | Tra.V | $\Delta$ |
| Multi-turn | $3.95_{\pm0.03}$ | $4.09_{\pm0.03}$ | 3.5% | $4.64_{\pm0.05}$ | $4.76_{\pm0.04}$ | 2.6% | $4.53_{\pm0.02}$ | $4.67_{\pm0.02}$ | 3.1% |
| Translation | $3.50_{\pm0.02}$ | $3.53_{\pm0.04}$ | 1.0% | $4.28_{\pm0.02}$ | $4.43_{\pm0.03}$ | 3.4% | $4.16_{\pm0.04}$ | $4.27_{\pm0.03}$ | 2.6% |
| Sum. | $3.66_{\pm0.02}$ | $3.76_{\pm0.03}$ | 2.6% | $4.51_{\pm0.02}$ | $4.64_{\pm0.02}$ | 2.7% | $4.32_{\pm0.03}$ | $4.46_{\pm0.03}$ | 3.1% |
| QA | $3.51_{\pm0.02}$ | $3.68_{\pm0.03}$ | 4.7% | $4.32_{\pm0.05}$ | $4.40_{\pm0.04}$ | 2.0% | $4.19_{\pm0.05}$ | $4.31_{\pm0.06}$ | 2.9% |
| Math | $4.61_{\pm0.05}$ | $4.70_{\pm0.03}$ | 1.8% | $5.37_{\pm0.03}$ | $5.39_{\pm0.05}$ | 0.4% | $5.13_{\pm0.01}$ | $5.21_{\pm0.02}$ | 1.5% |
| RAG | $4.05_{\pm0.04}$ | $4.17_{\pm0.05}$ | 3.1% | $4.63_{\pm0.02}$ | $4.76_{\pm0.06}$ | 2.8% | $4.60_{\pm0.03}$ | $4.68_{\pm0.04}$ | 1.7% |
| Avg. Accept. | $3.88_{\pm0.02}$ | $3.99_{\pm0.01}$ | 2.8% | $4.63_{\pm0.03}$ | $4.73_{\pm0.01}$ | 2.2% | $4.49_{\pm0.02}$ | $4.60_{\pm0.02}$ | 2.4% |
| Avg. Token/s | $51.2_{\pm1.2}$ | $52.5_{\pm1.1}$ | 2.5% | $54.0_{\pm0.6}$ | $54.9_{\pm1.2}$ | 1.7% | $57.3_{\pm1.3}$ | $58.5_{\pm0.8}$ | 2.1% |

Table 3: Acceptance length and throughput on Llama-68M with Llama2-7B.

| | Llama-68M (draft) & Llama2-7B (target) Temperature=1 | | | | | | | | |
| --- | --- | --- | --- | --- | --- | --- | --- | --- | --- |
| | Chain | | | Binary Tree | | | EAGLE Sparse Tree | | |
| Tasks | Tok.V | Tra.V | $\Delta$ | Tok.V | Tra.V | $\Delta$ | Tok.V | Tra.V | $\Delta$ |
| Multi-turn | $2.05_{\pm0.05}$ | $2.16_{\pm0.03}$ | 5.5% | $2.47_{\pm0.01}$ | $2.59_{\pm0.01}$ | 4.7% | $2.55_{\pm0.02}$ | $2.70_{\pm0.02}$ | 5.6% |
| Translation | $1.97_{\pm0.05}$ | $2.10_{\pm0.05}$ | 6.3% | $2.38_{\pm0.01}$ | $2.43_{\pm0.03}$ | 2.1% | $2.49_{\pm0.01}$ | $2.51_{\pm0.03}$ | 0.9% |
| Sum. | $1.77_{\pm0.04}$ | $1.86_{\pm0.05}$ | 4.9% | $2.14_{\pm0.01}$ | $2.27_{\pm0.03}$ | 5.8% | $2.25_{\pm0.02}$ | $2.36_{\pm0.02}$ | 4.7% |
| QA | $2.07_{\pm0.01}$ | $2.19_{\pm0.02}$ | 5.6% | $2.59_{\pm0.05}$ | $2.71_{\pm0.01}$ | 4.8% | $2.63_{\pm0.02}$ | $2.69_{\pm0.02}$ | 2.2% |
| Math | $2.01_{\pm0.05}$ | $2.15_{\pm0.04}$ | 7.0% | $2.49_{\pm0.05}$ | $2.67_{\pm0.06}$ | 7.0% | $2.57_{\pm0.02}$ | $2.72_{\pm0.01}$ | 6.0% |
| RAG | $2.09_{\pm0.05}$ | $2.19_{\pm0.03}$ | 4.8% | $2.56_{\pm0.05}$ | $2.69_{\pm0.05}$ | 5.0% | $2.63_{\pm0.02}$ | $2.71_{\pm0.06}$ | 3.2% |
| Avg. Accept. | $1.99_{\pm0.01}$ | $2.10_{\pm0.01}$ | 5.7% | $2.44_{\pm0.03}$ | $2.56_{\pm0.01}$ | 4.9% | $2.52_{\pm0.01}$ | $2.62_{\pm0.01}$ | 3.8% |
| Avg. Token/s | $58.0_{\pm0.7}$ | $60.8_{\pm0.8}$ | 4.8% | $59.4_{\pm0.8}$ | $61.6_{\pm0.6}$ | 3.7% | $69.1_{\pm0.9}$ | $71.2_{\pm1.0}$ | 3.0% |

generation speed without speculative decoding for Llama3.1-8B-Instruct is $34.5_{\pm0.1}$ token/s and for Llama2-7B is $37.3_{\pm0.1}$ token/s, and the speedup ratio can be calculated accordingly.

As can be observed from the results, compared with token-level verification, Traversal Verification achieves an average improvement in acceptance length of 2.2% to 5.7% across different tasks, tree architectures, and combinations of draft and target models. The performance gains from Traversal Verification exhibit variability depending on the specific configurations of draft and target models.

Since Traversal Verification operates through a bottom-up verification mechanism across the entire tree, it potentially introduces additional computational overhead compared to token-level verification. Consequently, the actual throughput improvement is slightly lower than the improvement in acceptance length. This issue can be mitigated through more optimized implementation.

### 4.3 Impact of Chain Depth and Tree Size

Since Traversal Verification considers the joint probability of the entire sequence, it is intuitive that the performance improvement will become more pronounced as the tree size and depth increase. To illustrate these effects, we perform experiments across varying chain depths and tree sizes. Specifically, for chain decoding, we conduct experiments at depths of 2, 4, 6, and 8. For tree decoding, we employ binary trees from depths of 2 to 5 (corresponding to trees with $2^3$-1, $2^4$-1, $2^5$-1, and $2^6$-1 nodes, respectively).

As shown in Figure 3, the advantage of Traversal Verification grows progressively with increasing chain depth and tree size. In specialized scenarios (*e.g.,* model offloading) where large tree sizes are permissible (for example, Sequoia [4] utilizes trees with 768 nodes and depth exceeding 20), Traversal Verification is expected to demonstrate even greater performance gains.

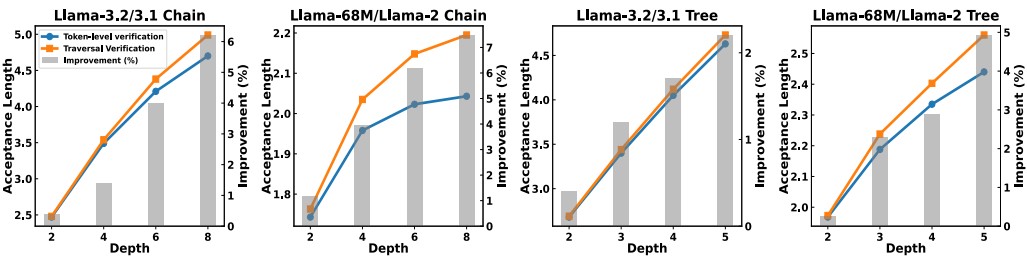

Figure 3: Acceptance lengths and improvements under different chain depths and tree sizes.

## 4.4 Impact of Temperature

We investigate the impact of temperature on Traversal Verification. Intuitively, as the temperature decreases (*i.e.,* the probability distribution becomes more concentrated), the performance gap between token-level verification and Traversal Verification narrows. Conversely, at higher temperatures, Traversal Verification demonstrates more pronounced advantages.

Table 4: Acceptance lengths under different temperature.

| Temp. | Chain | | | Binary Tree | | | EAGLE Sparse Tree | | |
|---|---|---|---|---|---|---|---|---|---|
| | Tok.V | Tra.V | $\Delta$ | Tok.V | Tra.V | $\Delta$ | Tok.V | Tra.V | $\Delta$ |
| 0.2 | $4.16_{\pm 0.01}$ | $4.20_{\pm 0.01}$ | 1.0% | $5.01_{\pm 0.02}$ | $5.07_{\pm 0.02}$ | 1.2% | $4.77_{\pm 0.03}$ | $4.84_{\pm 0.01}$ | 1.5% |
| 0.4 | $4.14_{\pm 0.02}$ | $4.20_{\pm 0.02}$ | 1.4% | $5.00_{\pm 0.01}$ | $5.06_{\pm 0.01}$ | 1.2% | $4.76_{\pm 0.02}$ | $4.83_{\pm 0.02}$ | 1.5% |
| 0.6 | $4.11_{\pm 0.02}$ | $4.17_{\pm 0.03}$ | 1.5% | $4.92_{\pm 0.03}$ | $5.00_{\pm 0.01}$ | 1.5% | $4.71_{\pm 0.01}$ | $4.78_{\pm 0.01}$ | 1.5% |
| 0.8 | $4.02_{\pm 0.02}$ | $4.11_{\pm 0.01}$ | 2.2% | $4.81_{\pm 0.02}$ | $4.90_{\pm 0.02}$ | 1.7% | $4.64_{\pm 0.02}$ | $4.72_{\pm 0.01}$ | 1.7% |
| 1.0 | $3.88_{\pm 0.02}$ | $3.99_{\pm 0.01}$ | 2.8% | $4.63_{\pm 0.03}$ | $4.73_{\pm 0.01}$ | 2.2% | $4.49_{\pm 0.02}$ | $4.60_{\pm 0.02}$ | 2.4% |

Table 4 presents the acceptance length of Traversal Verification and token-level verification across different temperature settings, using Llama3.2-1B-Instruct and Llama3.1-8B-Instruct as the draft and target models, respectively. The depths of chain and binary tree are set to 5. The superiority of Traversal Verification increases with rising temperature, aligning with our intuitive expectations. It is worth noting that Llama2-7B may generate repeated tokens at lower temperatures, leading to unreliable acceptance length measurements; therefore, we omit the results for Llama2 in this analysis.

## 5 Related work

Significant efforts have been devoted to accelerating LLMs. Some approaches directly reduce memory access and computational costs through techniques such as quantization [8, 9, 33, 22] and knowledge distillation [11, 17, 37]. Some other works focus on architectural innovations, such as Mixture of Experts (MoE) [15, 7], where only a subset of model parameters is activated during inference, thereby improving inference speed.

Speculative decoding [3, 19] introduces a distinct drafting-verification paradigm that leaves the LLM itself unchanged. Researches on speculative decoding primarily focus on two directions. 1) Better alignment between the draft and the target model, such as EAGLE [21, 20] and Medusa [2] series. 2) Better verification strategies, such as innovations in tree structures [20, 4, 31] and verification algorithms, which are more closely related to this work.

In chain decoding scenarios, Block Verification [27] and Asps [12] identify the sub-optimality in token-level verification and propose enhancements. SpecTr [28] extends chain decoding to multi-candidate settings by formulating it as an optimal transport problem solved via linear programming, while SpecInfer [23] employs Recursive Rejection Sampling for multi-candidate situations. Subsequent works refine this approach into RRSw (recursive rejection sampling without replacement) [4, 21, 14, 35], preventing repeated sampling and rejection of identical tokens, thereby improving acceptance rates. Beyond standard sampling, SpecHub [26] and Greedy Sampling [13] adopt hybrid strategies: deterministically selecting top-K candidates with the highest probability and sampling other candidates probabilistically, achieving higher acceptance rates in specific scenarios.

# 6 Conclusion

This paper proposes Traversal Verification, a novel speculative decoding algorithm that significantly enhances the acceptance length, thereby improving the throughput of LLM inference. We rethink the limitations of existing token-level verification methods and adopt a bottom-up verification strategy that allows sequence-level acceptance and full utilization of drafted tokens. We theoretically prove the losslessness of Traversal Verification and its optimality when the decoding tree degenerates into a single chain. Experimental results show that Traversal Verification consistently improves the acceptance length and throughput of over existing speculative tree decoding methods across various tasks, tree structures, and combinations of draft and target models.

## Acknowledgments and Disclosure of Funding

This project is fully funded by Lenovo. We would like to express special thanks to the Lenovo AI Lab and the Lenovo Model Factory Team for their valuable support in providing computing resources.

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

# A  Formal Proof of Losslessness of Traversal Verification

We first prove a necessary and sufficient condition for a valid tree verification algorithm (Definition 3.2). Our proof technique is analogous to [27, Lemma 2 in Appendix.B] and extends the original lemma to a tree-structured case.

**Lemma A.1.** $\forall \mathcal{M}_s, \mathcal{M}_b$, let $T$ be a decoding tree rooted at $X_0$ base on $\mathcal{M}_s$, and $\gamma_{\max} := \max_{v \subset T} \gamma_v$ be the maximum depth along all chains in $T$. The output of a tree verification algorithm $\mathcal{A}_{\mathrm{ver}}$ is denoted as

$$(X^\tau, Y) = \mathcal{A}_{\mathrm{ver}}(T, \mathcal{M}_s, \mathcal{M}_b).$$

Let $Z^{\gamma_{\max}} = (Z_0, Z_1, \ldots, Z_{\gamma_{\max}})$ be a sequence defined as follows:

$$Z^{\gamma_{\max}} = \begin{cases} X^\tau, & \tau = \gamma_{\max} \\ (X^\tau, Y, Z_{>\tau+1}), & \tau < \gamma_{\max} \end{cases},$$

with $Z_{>\tau+1} := (Z_{\tau+2}, \ldots, Z_{\gamma_{\max}})$ generated from $\mathcal{M}_b(\cdot|X^\tau, Y)$. Then the tree verification algorithm $\mathcal{A}_{\mathrm{ver}}$ is valid if and only if

$$Z^{\gamma_{\max}} \sim \mathcal{M}_b(\cdot|X_0). \tag{5}$$

*Proof.* We first prove the sufficiency, i.e., Equation (5) implies that $\mathcal{A}_{\mathrm{ver}}$ satisfies Definition 3.2.

Taking the output $(X^\tau, Y)$ as a new prefix into $\mathcal{A}_{\mathrm{ver}}$, we obtain

$$(\widetilde{X}^{\widetilde{\tau}}, \widetilde{Y}) = \mathcal{A}_{\mathrm{ver}}(\widetilde{T}, \mathcal{M}_s, \mathcal{M}_b),$$

with the root of $\widetilde{T}$ being $\widetilde{X}_0 = (X^\tau, Y)$ and then generate

$$\widetilde{Z}^{\gamma_{\max}} = \begin{cases} \widetilde{X}^{\widetilde{\tau}}, & \widetilde{\tau} = \gamma_{\max} \\ (\widetilde{X}^{\widetilde{\tau}}, \widetilde{Y}, \widetilde{Z}_{>\widetilde{\tau}+1}), & \widetilde{\tau} < \gamma_{\max} \end{cases}$$

with $\widetilde{Z}_{>\widetilde{\tau}+1} \sim \mathcal{M}_b(\cdot|\widetilde{X}^{\widetilde{\tau}}, \widetilde{Y})$. Note that by Equation (5), we have

$$\widetilde{Z}^{\gamma_{\max}} \sim \mathbb{P}(X^\tau, Y)\mathcal{M}_b(\cdot|\widetilde{X}_0),$$

and

$$Z^{\gamma_{\max}}, E^* \sim \mathcal{M}_b(\cdot|X_0),$$

Here, $E^*$ is an extension sequence of $Z^{\gamma_{\max}}$ generated from $\mathcal{M}_b(E^*|Z^{\gamma_{\max}})$, such that the combined sequence $(Z^{\gamma_{\max}}, E^*)$ has the same number of tokens as $\widetilde{Z}^{\gamma_{\max}}$. For the sequences $(Z^{\gamma_{\max}}, E^*)$ and $\widetilde{Z}^{\gamma_{\max}}$, by taking the expectation over all the random variables after $(X^\tau, Y)$, we get

$$\mathbb{P}(X^\tau, Y) = \mathcal{M}_b(X^\tau, Y|X_0),$$

namely, the proof of the sufficiency is completed.

The necessity is straightforward: If $\tau < \gamma_{\max}$, Equation (5) holds trivially. If $\tau = \gamma_{\max}$, then $Z^{\gamma_{\max}} = X^\tau$ and $Y \sim \mathcal{M}_b(\cdot|X^\tau)$. By (4) in Definition 3.2, we also have

$$Z^{\gamma_{\max}} \sim \mathcal{M}_b(\cdot|X_0).$$

In conclusion, the proof of the necessity is also completed. □

## A.1  Proof of Theorem 3.3

By Lemma A.1, it would be enough to prove that Traversal Verification satisfies Equation (5). We observe the inherent *self-similar* property of Traversal Verification: When arbitrarily selecting a parent node and rejecting it, the algorithm has already evaluated all its descendant nodes through the same traversal mechanism. In other words, Traversal Verification effectively applies a recursive instance of itself to the local subtree rooted at the current parent node. Leveraging this self-similar property, we establish the following stronger lemma than Theorem 3.3.

**Lemma A.2.** $\forall \mathcal{M}_s, \mathcal{M}_b$, let $T$ be a decoding tree rooted at $X_0$ base on $\mathcal{M}_s$ and $\gamma_{\max} := \max_{v \subset T} \gamma_v$ be the maximum depth along all chains in $T$. The first chain in $T$ is denoted as $\alpha = (\alpha_0, \alpha_1, \ldots, \alpha_{\gamma_\alpha})$ from root $\alpha_0 = X_0$ to the first leaf node $\alpha_{\gamma_\alpha}$. $Z^{\gamma_{\max}}$ is the sequence generated by Traversal Verification $\mathcal{A}_{\mathrm{tra}}(T, \mathcal{M}_s, \mathcal{M}_b)$ (i.e., Algorithm 3) in Lemma A.1. Then the following statements hold, $\forall 0 \leqslant \ell \leqslant \gamma_\alpha$,

$$\mathbb{P}(Z^\ell = \alpha^\ell) = p_\alpha^{ini}(\alpha_\ell) \quad \text{and} \quad \mathbb{P}(Z^{\gamma_{\max}} = (\alpha^\ell, Z_{>\ell})) = p_\alpha^{ini}(\alpha_\ell)\mathcal{M}_b(Z_{>\ell}|\alpha^\ell). \tag{6}$$

*Proof.* When $\gamma_\alpha = 0$, i.e., the tree $T$ contains only the root node $X_0$, then $\gamma_{\max} = 0$, $Z^{\gamma_{\max}} = X_0$ and the conclusion (6) holds trivially. Therefore, in subsequent proofs, we only need to consider the case where $\gamma_\alpha \geqslant 1$.

Next, we begin to prove the statements in (6) hold for any fixed $0 \leqslant \ell \leqslant \gamma_\alpha$ by induction on the number of descendant nodes of $\alpha_\ell$. For simplicity, , we collect all the children nodes of $\alpha_\ell$ as a new set $C(\alpha_\ell) \subset T$ and all the descendant nodes of $\alpha_\ell$ as $D(\alpha_\ell) \subset T$.

When the number $|D(\alpha_\ell)| = 0$, i.e., $\alpha_\ell$ is the leave node of the first chain $\alpha$, then $\ell = \gamma_\alpha$ and $Z^{\gamma_\alpha} = \alpha^{\gamma_\alpha}$ means that the traversal algorithm $\mathcal{A}_{\mathrm{tra}}$ accepts the first chain $\alpha$ directly. Thus,

$$\mathbb{P}(Z^{\gamma_\alpha} = \alpha^{\gamma_\alpha}) = p_\alpha^{ini}(\alpha_{\gamma_\alpha}) \quad \text{and} \quad \mathbb{P}(Z^{\gamma_{\max}} = (\alpha^{\gamma_\alpha}, Z_{>\gamma_\alpha})) = p_\alpha^{ini}(\alpha_{\gamma_\alpha}) \mathcal{M}_b(Z_{>\gamma_\alpha}|\alpha^\ell).$$

Suppose the two equations in (6) hold when $|D(\alpha_\ell)| \leqslant k$. Then when $|D(\alpha_\ell)| = k+1$, we know $D(\alpha_\ell)$ is nonempty since the node $\alpha_{\ell+1} \in D(\alpha_\ell)$. Trivially, we have $|D(\alpha_{\ell+1})| \leqslant k$, then by the induction hypothesis,

$$\mathbb{P}(Z^{\ell+1} = \alpha^{\ell+1}) = p_\alpha^{ini}(\alpha_{\ell+1}) \tag{7}$$

$$\mathbb{P}(Z^{\gamma_{\max}} = (\alpha^{\ell+1}, Z_{>\ell+1})) = p_\alpha^{ini}(\alpha_{\ell+1}) \mathcal{M}_b(Z_{>\ell+1}|\alpha^{\ell+1}). \tag{8}$$

For Traversal Verification $\mathcal{A}_{\mathrm{tra}}(T, \mathcal{M}_s, \mathcal{M}_b)$, the probability

$$\begin{aligned} \mathbb{P}(Z^\ell = \alpha^\ell) &= \mathbb{P}(Z_{\ell+1} = \alpha_{\ell+1}, Z^\ell = \alpha^\ell) + \mathbb{P}(Z_{\ell+1} \neq \alpha_{\ell+1}, Z^\ell = \alpha^\ell) \\ &= p_\alpha^{ini}(\alpha_{\ell+1}) + \mathbb{P}(Z_{\ell+1} \neq \alpha_{\ell+1}) \cdot \mathbb{P}(Z^\ell = \alpha^\ell | Z_{\ell+1} \neq \alpha_{\ell+1}) \\ &= p_\alpha^{ini}(\alpha_{\ell+1}) + (1 - p_\alpha^{ini}(\alpha_{\ell+1})) \cdot \mathbb{P}(Z^\ell = \alpha^\ell | Z_{\ell+1} \neq \alpha_{\ell+1}) \end{aligned} \tag{9}$$

In the case of $Z_{\ell+1} \neq \alpha_{\ell+1}$, namely, all the nodes in $D(\alpha_{\ell+1}) \cup \{\alpha_{\ell+1}\}$ have been removed from the original tree $T$, the remaining tree $T_{\mathrm{new}} := T - D(\alpha_{\ell+1}) - \{\alpha_{\ell+1}\}$ modifies only the following parameters compared to the original tree:

- the acceptance rate $p_\alpha'(\alpha_\ell)$:

$$p_\alpha'(\alpha_\ell) = \frac{\sum_x [p_\alpha^{ini}(\alpha_\ell) \cdot \mathcal{M}_b(x|\alpha^\ell) - \mathcal{M}_s(x|\alpha^\ell)]_+}{\sum_x [p_\alpha^{ini}(\alpha_\ell) \cdot \mathcal{M}_b(x|\alpha^\ell) - \mathcal{M}_s(x|\alpha^\ell)]_+ + 1 - p_\alpha^{ini}(\alpha_\ell)}. \tag{10}$$

- the distributions $\mathcal{M}_b'(x|\alpha^\ell)$ and $\mathcal{M}_s'(x|\alpha^\ell)$ for all *children nodes* of $\alpha_\ell$:

$$\mathcal{M}_b'(x|\alpha^\ell) = \frac{[p_\alpha^{ini}(\alpha_\ell) \cdot \mathcal{M}_b(x|\alpha^\ell) - \mathcal{M}_s(x|\alpha^\ell)]_+}{\sum_x [p_\alpha^{ini}(\alpha_\ell) \cdot \mathcal{M}_b(x|\alpha^\ell) - \mathcal{M}_s(x|\alpha^\ell)]_+}, \quad \forall x \in \mathcal{X}, \tag{11}$$

$$\mathcal{M}_s'(\alpha_{\ell+1}|\alpha^\ell) = 0 \text{ and } \mathcal{M}_s'(x|\alpha^\ell) = \frac{\mathcal{M}_s(x|\alpha^\ell)}{1 - \mathcal{M}_s(\alpha_{\ell+1}|\alpha^\ell)} \quad \forall x \neq \alpha_{\ell+1}. \tag{12}$$

Therefore, after $\alpha_{\ell+1}$ has been rejected, the acceptance rate of the parent node $\alpha_\ell$ decreases from $p_\alpha^{ini}(\alpha_\ell)$ to $p_\alpha'(\alpha_\ell)$, and the remaining children nodes of $\alpha_\ell$ in $T_{\mathrm{new}}$ are stochastic sampling nodes on $\mathcal{M}_s'(\cdot|\alpha^\ell)$, with their corresponding target distributions being $\mathcal{M}_b'(\cdot|\alpha^\ell)$. By the *self-similar* property of $\mathcal{A}_{\mathrm{tra}}$, we observe that in the remaining tree $T_{\mathrm{new}}$, Traversal Verification utilizes only the acceptance probability $p'(\alpha_\ell)$ of parent node $\alpha_\ell$, the new distributions $\mathcal{M}_s'(\cdot|\alpha^\ell)$, $\mathcal{M}_b'(\cdot|\alpha^\ell)$ of children nodes $C(\alpha_\ell)$, and the original distributions $\mathcal{M}_s(\cdot|\alpha^\ell)$ and $\mathcal{M}_b(\cdot|\alpha^\ell)$ of other descendant nodes $D(\alpha_\ell) - C(\alpha_\ell)$. Since $\alpha_{\ell+1} \notin T_{\mathrm{new}}$, the number of descendant nodes of $\alpha_\ell$ in new tree $T_{\mathrm{new}}$ is less than the original $|D(\alpha_\ell)|$, by the induction hypothesis, we know

$$\begin{aligned} \mathbb{P}(Z^\ell = \alpha^\ell | Z_{\ell+1} \neq \alpha_{\ell+1}) &= \mathbb{P}(\text{accept } \alpha_\ell | \text{reject } \alpha_{\ell+1}) \\ &= \mathbb{P}(\text{accept } \alpha_\ell \text{ in } T_{\mathrm{new}}) = p_\alpha'(\alpha_\ell), \end{aligned} \tag{13}$$

$$\begin{aligned} \mathbb{P}(Z^{\gamma_{\max}} = (\alpha^\ell, Z_{>\ell}) | Z_{\ell+1} \neq \alpha_{\ell+1}) &= \mathbb{P}(Z^{\gamma_{\max}} = (\alpha^\ell, Z_{>\ell}) | T_{\mathrm{new}}) \\ &= p_\alpha'(\alpha_\ell) \mathcal{M}_b'(Z_{\ell+1}|\alpha^\ell) \mathcal{M}_b(Z_{>\ell+1}|\alpha^\ell, Z_{\ell+1}). \end{aligned} \tag{14}$$

Now, we begin to prove $\mathbb{P}(Z^\ell = \alpha^\ell) = p_\alpha^{ini}(\alpha_\ell)$ at first.

$$\begin{aligned} \mathbb{P}(Z^\ell = \alpha^\ell) &\overset{(9)}{=} p_\alpha^{ini}(\alpha_{\ell+1}) + (1 - p_\alpha^{ini}(\alpha_{\ell+1})) \cdot \mathbb{P}(Z^\ell = \alpha^\ell | Z_{\ell+1} \neq \alpha_{\ell+1}) \\ &\overset{(13)}{=} p_\alpha^{ini}(\alpha_{\ell+1}) + (1 - p_\alpha^{ini}(\alpha_{\ell+1})) \cdot p_\alpha'(\alpha_\ell). \end{aligned} \tag{15}$$

Since $\mathbb{P}(Z^\ell = \alpha^\ell)$ is independent to the random variable $\alpha_{\ell+1}$, we have

$$\mathbb{P}(Z^\ell = \alpha^\ell)$$
$$= \mathbb{E}_{\alpha_{\ell+1}}[\mathbb{P}(Z^\ell = \alpha^\ell)]$$
$$= \sum_x p_\alpha^{ini}(\alpha_{\ell+1} = x)\mathcal{M}_s(x|\alpha^\ell) + \left[1 - \sum_x p_\alpha^{ini}(\alpha_{\ell+1} = x)\mathcal{M}_s(x|\alpha^\ell)\right] \cdot p_\alpha'(\alpha_\ell)$$
$$= \sum_x \min\left\{p_\alpha^{ini}(\alpha_\ell)\mathcal{M}_b(x|\alpha^\ell), \mathcal{M}_s(x|\alpha^\ell)\right\} + p_\alpha'(\alpha_\ell)\sum_x \left[\mathcal{M}_s(x|\alpha^\ell) - p_\alpha^{ini}(\alpha_\ell)\mathcal{M}_b(x|\alpha^\ell)\right]_+$$
$$\overset{(10)}{=} \sum_x \min\left\{p_\alpha^{ini}(\alpha_\ell)\mathcal{M}_b(x|\alpha^\ell), \mathcal{M}_s(x|\alpha^\ell)\right\} + \sum_x \left[p_\alpha^{ini}(\alpha_\ell)\mathcal{M}_b(x|\alpha^\ell) - \mathcal{M}_s(x|\alpha^\ell)\right]_+$$
$$= \sum_x \left[p_\alpha^{ini}(\alpha_\ell)\mathcal{M}_b(x|\alpha^\ell) + \mathcal{M}_s(x|\alpha^\ell)\right] - \sum_x \mathcal{M}_s(x|\alpha^\ell)$$
$$= p_\alpha^{ini}(\alpha_\ell).$$

Then we begin to prove the second statement of (6), i.e.,

$$\mathbb{P}(Z^{\gamma_{\max}} = (\alpha^\ell, Z_{>\ell})) = p_\alpha^{ini}(\alpha_\ell)\mathcal{M}_b(Z_{>\ell}|\alpha^\ell). \tag{16}$$

For any sequence $x^{\gamma_{\max}}$ satisfying $x^\ell = \alpha^\ell$, we have

$$\mathbb{P}(Z^{\gamma_{\max}} = x^{\gamma_{\max}})$$
$$= \mathbb{P}(\alpha_{\ell+1} = x_{\ell+1})\mathbb{P}(Z^{\gamma_{\max}} = (\alpha^{\ell+1}, x_{>\ell+1})|\alpha_{\ell+1} = x_{\ell+1})$$
$$\quad + \sum_{x' \neq x_{\ell+1}} \mathbb{P}(\alpha_{\ell+1} = x')\mathbb{P}(\text{reject } \alpha_{\ell+1})\mathbb{P}(Z^{\gamma_{\max}} = (\alpha^\ell, x_{>\ell})|\text{reject } \alpha_{\ell+1})$$
$$\overset{(8)}{=} \mathcal{M}_s(x_{\ell+1}|\alpha^\ell) \cdot p_\alpha^{ini}(\alpha_{\ell+1} = x_{\ell+1})\mathcal{M}_b(x_{>\ell+1}|\alpha^\ell, x_{\ell+1})$$
$$\quad + \sum_{x' \neq x_{\ell+1}} \mathcal{M}_s(x'|\alpha^\ell)[1 - p_\alpha^{ini}(\alpha_{\ell+1} = x')]\mathbb{P}(Z^{\gamma_{\max}} = (\alpha^\ell, x_{>\ell})|x_{\ell+1} \neq \alpha_{\ell+1})$$
$$\overset{(14)}{=} \mathcal{M}_s(x_{\ell+1}|\alpha^\ell) \cdot p_\alpha^{ini}(\alpha_{\ell+1} = x_{\ell+1})\mathcal{M}_b(x_{>\ell+1}|\alpha^\ell, x_{\ell+1})$$
$$\quad + p_\alpha'(\alpha_\ell)\mathcal{M}_b'(x_{\ell+1}|\alpha^\ell)\mathcal{M}_b(x_{>\ell+1}|\alpha^\ell, x_{\ell+1})\sum_{x' \neq x_{\ell+1}} \mathcal{M}_s(x'|\alpha^\ell)[1 - p_\alpha^{ini}(\alpha_{\ell+1} = x')]$$
$$= \mathcal{M}_b(x_{>\ell+1}|\alpha^\ell, x_{\ell+1}) \cdot \min\left\{p_\alpha^{ini}(\alpha_\ell)\mathcal{M}_b(x_{\ell+1}|\alpha^\ell), \mathcal{M}_s(x_{\ell+1}|\alpha^\ell)\right\}$$
$$\quad + \mathcal{M}_b(x_{>\ell+1}|\alpha^\ell, x_{\ell+1}) \cdot p_\alpha'(\alpha_\ell)\mathcal{M}_b'(x_{\ell+1}|\alpha^\ell)\sum_{x' \neq x_{\ell+1}} \left[\mathcal{M}_s(x'|\alpha^\ell) - p_\alpha^{ini}(\alpha_\ell)\mathcal{M}_b(x'|\alpha^\ell)\right]_+$$
$$\tag{$*$}$$

We evaluate the value of Equation $(*)$ via case analysis of $x_{\ell+1}$:

- Case 1: $p_\alpha^{ini}(\alpha_\ell)\mathcal{M}_b(x_{\ell+1}|\alpha^\ell) \leqslant \mathcal{M}_s(x_{\ell+1}|\alpha^\ell)$.

  Since $\mathcal{M}_b'(x_{\ell+1}|\alpha^\ell) = 0$ in this case (see (11)), the Equation $(*)$ is equal to

  $$(*) = \mathcal{M}_b(x_{>\ell+1}|\alpha^\ell, x_{\ell+1}) \cdot p_\alpha^{ini}(\alpha_\ell)\mathcal{M}_b(x_{\ell+1}|\alpha^\ell) = p_\alpha^{ini}(\alpha_\ell)\mathcal{M}_b(x_{>\ell}|\alpha^\ell).$$

- Case 2: $p_\alpha^{ini}(\alpha_\ell)\mathcal{M}_b(x_{\ell+1}|\alpha^\ell) > \mathcal{M}_s(x_{\ell+1}|\alpha^\ell)$.

  Since $[\mathcal{M}_s(x_{\ell+1}|\alpha^\ell) - p_\alpha^{ini}(\alpha_\ell)\mathcal{M}_b(x_{\ell+1}|\alpha^\ell)]_+ = 0$, we have

  $$\sum_{x' \neq x_{\ell+1}} \left[\mathcal{M}_s(x'|\alpha^\ell) - p_\alpha^{ini}(\alpha_\ell)\mathcal{M}_b(x'|\alpha^\ell)\right]_+$$
  $$= \sum_{x'} \left[\mathcal{M}_s(x'|\alpha^\ell) - p_\alpha^{ini}(\alpha_\ell)\mathcal{M}_b(x'|\alpha^\ell)\right]_+.$$

Together with (10) and (11), we get

$$p'_\alpha(\alpha_\ell)\mathcal{M}'_b(x_{\ell+1}|\alpha^\ell) \sum_{x'\neq x_{\ell+1}} \left[\mathcal{M}_s(x'|\alpha^\ell) - p^{ini}_\alpha(\alpha_\ell)\mathcal{M}_b(x'|\alpha^\ell)\right]_+$$

$$= p'_\alpha(\alpha_\ell)\mathcal{M}'_b(x_{\ell+1}|\alpha^\ell)\left\{\sum_{x'}\left[p^{ini}_\alpha(\alpha_\ell)\mathcal{M}_b(x'|\alpha^\ell) - \mathcal{M}_s(x'|\alpha^\ell)\right]_+ + 1 - p^{ini}_\alpha(\alpha_\ell)\right\}$$

$$= [p^{ini}_\alpha(\alpha_\ell)\mathcal{M}_b(x_{\ell+1}|\alpha^\ell) - \mathcal{M}_s(x_{\ell+1}|\alpha^\ell)]_+$$

$$= p^{ini}_\alpha(\alpha_\ell)\mathcal{M}_b(x_{\ell+1}|\alpha^\ell) - \mathcal{M}_s(x_{\ell+1}|\alpha^\ell).$$

Taking it into $(*)$, then

$$(*) = \mathcal{M}_b(x_{>\ell+1}|\alpha^\ell, x_{\ell+1})\left(\mathcal{M}_s(x_{\ell+1}|\alpha^\ell) + p^{ini}_\alpha(\alpha_\ell)\mathcal{M}_b(x_{\ell+1}|\alpha^\ell) - \mathcal{M}_s(x_{\ell+1}|\alpha^\ell)\right)$$

$$= \mathcal{M}_b(x_{>\ell+1}|\alpha^\ell, x_{\ell+1})p^{ini}_\alpha(\alpha_\ell)\mathcal{M}_b(x_{\ell+1}|\alpha^\ell) = p^{ini}_\alpha(\alpha_\ell)\mathcal{M}_b(x_{>\ell}|\alpha^\ell).$$

In conclusion, $(*) = p^{ini}_\alpha(\alpha_\ell)\mathcal{M}_b(x_{>\ell}|\alpha^\ell)$, i.e.,

$$\mathbb{P}(Z^{\gamma\max} = x^{\gamma\max}) = p^{ini}_\alpha(\alpha_\ell)\mathcal{M}_b(x_{>\ell}|\alpha^\ell), \quad \forall x^{\gamma\max} = (\alpha^\ell, x_{>\ell}).$$

Thus, the Equation (16) holds and we have proven by mathematical induction that, in Traversal Verification $\mathcal{A}_{\mathrm{tra}}$, for any node $\alpha_\ell$ in the initial first chain, the following equations hold:

$$\mathbb{P}(Z^\ell = \alpha^\ell) = p^{ini}_\alpha(\alpha_\ell) \quad \text{and} \quad \mathbb{P}(Z^{\gamma\max} = (\alpha^\ell, Z_{>\ell})) = p^{ini}_\alpha(\alpha_\ell)\mathcal{M}_b(Z_{>\ell}|\alpha^\ell).$$

$\square$

Theorem 3.3 can be directly deduced from this lemma. Specifically, by setting $\ell = 0$ in Lemma A.2, we immediately obtain that

$$\mathbb{P}(Z^{\gamma\max} = (X_0, Z_{>0})) = \mathcal{M}_b(Z_{>0}|X_0).$$

Since $\mathcal{M}_b(X_0|X_0) = 1$, we know

$$Z^{\gamma\max} \sim \mathcal{M}_b(\cdot|X_0).$$

Therefore, the proof of and Theorem 3.3 has been completed.

## B    Formal Proof of Single-chain Optimality

To establish the optimality of Traversal Verification in the single-chain case, we need to introduce two lemmas presented in [27].

**Lemma B.1** (Lemma 3 in [27]). *Let $T = (\alpha_0, \ldots, \alpha_\gamma)$ be a decoding chain based on $\mathcal{M}_s$, $\mathcal{A}_{block}$ be Block Verification proposed in [27], and*

$$(X^\tau, Y) = \mathcal{A}_{\mathrm{block}}(T, \mathcal{M}_s, \mathcal{M}_b).$$

*Then we have $\forall \ell \leqslant \gamma$,*

$$\mathbb{P}(\tau \geqslant \ell|X^\ell = \alpha^\ell) = p^{ini}_\alpha(\alpha_\ell).$$

**Lemma B.2** (Lemma 4 in [27]). *For chain verification algorithms that satisfy the constraints in Lemma A.2, we have $\forall \ell \leqslant \gamma$,*

$$\mathbb{P}(\tau \geqslant \ell|X^\ell = \alpha^\ell) \leqslant p^{ini}_\alpha(\alpha_\ell).$$

*Proof.* It suffices to observe that when the stochastic sampling tree reduces to a single-chain structure, the equivalent definition Lemma A.1 of *valid chain verification algorithm* in this paper is identical to the equivalent definition [27, Lemma 2] of the *valid draft verification algorithm*. Therefore, Lemmas B.1 and B.2 hold automatically as the directly applications of [27, Lemma 3 and Lemma 4].    $\square$

Note that when the sampling tree reduces to one single chain, Lemma A.2 shows that the probability of Traversal Verification accepting at least $\ell$ tokens is

$$\mathbb{P}(\tau \geqslant \ell|X^\ell = \alpha^\ell) = \mathbb{P}(Z^\ell = \alpha^\ell) = p^{ini}_\alpha(\alpha_\ell).$$

By Lemmas B.1 and B.2, we show that among all valid chain verification algorithms (i.e., valid draft verification algorithms satisfying the constraints in [27, Lemma 2]), Traversal Verification accepts any given subsequence with the highest probability as the same as Block Verification. Specifically, for a given decoding chain $\alpha = (\alpha_0, \ldots, \alpha_\gamma)$ based on $\mathcal{M}_s$, we have

$$\mathbb{E}[N_{\text{traversal}}] = \mathbb{E}_{\alpha \sim \mathcal{M}_s}\left[\sum_\ell \mathbb{P}(\tau_{\text{traversal}} \geqslant \ell | X^\ell = \alpha^\ell)\right] = \mathbb{E}_{\alpha \sim \mathcal{M}_s}\left[\sum_\ell p_\alpha^{ini}(\alpha_\ell)\right],$$

$$\mathbb{E}[N_{\text{block}}] = \mathbb{E}_{\alpha \sim \mathcal{M}_s}\left[\sum_\ell \mathbb{P}(\tau_{\text{block}} \geqslant \ell | X^\ell = \alpha^\ell)\right] = \mathbb{E}_{\alpha \sim \mathcal{M}_s}\left[\sum_\ell p_\alpha^{ini}(\alpha_\ell)\right],$$

$$\mathbb{E}[N_{\text{verify}}] = \mathbb{E}_{\alpha \sim \mathcal{M}_s}\left[\sum_\ell \mathbb{P}(\tau_{\text{verify}} \geqslant \ell | X^\ell = \alpha^\ell)\right] \leqslant \mathbb{E}_{\alpha \sim \mathcal{M}_s}\left[\sum_\ell p_\alpha^{ini}(\alpha_\ell)\right].$$

This implies Theorem 3.4 holds.

## C   Evaluation of Generation Quality

Although we have already provided a mathematically rigorous proof for the losslessness of Traversal Verification, we understand that it is also important to present experimental results regarding generation quality. We would like to emphasize that the primary application scenario of Traversal Verification lies in non-greedy generation, therefore, due to the randomness introduced by sampling and hardware fluctuations, there will be variations in the results generated each time. Consequently, the losslessness of Traversal Verification cannot be "proven" through experiments, and the measurement of generation quality serves merely as a reference.

We follow the method used in Medusa [2] for measuring generation quality: we use the MT-Bench [36] dataset and employ a state-of-the-art LLM as a judge to evaluate the quality of generation.

Table 5 presents the evaluation of generation quality with Llama3.1-8B-Instruct (using the same 10-point scale as Medusa, higher score is better). We use Gemini-2.5-Flash [6] as the judge model to assess the quality of the MT-bench responses. For all experiments, we ran them three times and report the average.

Table 5: Evaluation of generation quality.

| Method | Verification Strategy | Quality |
|---|---|---|
| Autoregressive | N/A | 6.72 |
| Chain | Token-level | 6.78 |
| | Traversal | 6.77 |
| EAGLE Sparse Tree | Token-level | 6.76 |
| | Traversal | 6.79 |
| Binary Tree | Token-level | 6.69 |
| | Traversal | 6.74 |

The results show that Traversal Verification maintains roughly the same generation quality as both naive generation and token-level verification, which serves as evidence for its lossless property.

## D   Statistical Methods and Additional Results

When calculating the acceptance length, the results may vary slightly due to different statistical methods. Specifically, the default statistical method of Spec-Bench can generally be described as "the average tokens generated per drafting-verification cycle across the whole dataset".

However, this statistical method is not entirely appropriate. Because Spec-Bench covers diverse tasks, the answer length for each task and each sample can vary significantly. For instance, text generation tasks (such as "Compose an engaging travel blog post about a recent trip to Hawaii") often have

longer responses than short translation queries (such as "Translate German to English: Dennoch : Die Wahrheit auszusprechen ist kein Verbrechen"). Calculating the average acceptance length by aggregating all generated tokens will clearly be heavily influenced by long responses. Therefore, when we compute the average acceptance length, we calculate it for *each item* first and then take the average across all items. This introduces a slight difference from the default metric used in Spec-Bench.

We provide the acceptance lengths obtained using different statistical methods in Table 6. We also include the speedup ratios. To align with the official Spec-Bench benchmark results, we use EAGLE-Vicuna-7B-v1.3 [21] as the draft model and Vicuna-7B-v1.3 [36] as the target model. As shown, although the acceptance length slightly varies under different statistical methods, Traversal Verification consistently achieves a stable improvement.

Table 6: Comparison of acceptance lengths using different statistical methods.

| Tree Structure | Verification | Acceptance length | | Speedup |
| | | default (by token) | ours (by item) | |
| --- | --- | --- | --- | --- |
| Chain | Token-level | 2.57 | 2.51 | 1.77x |
| | Traversal | **2.63** | **2.57** | **1.81x** |
| Binary Tree | Token-level | 3.11 | 3.04 | 1.87x |
| | Traversal | **3.22** | **3.12** | **1.92x** |
| EAGLE Sparse Tree | Token-level | 3.18 | 3.10 | 2.00x |
| | Traversal | **3.26** | **3.16** | **2.04x** |

# E   Traversal Order

After the tree structure was determined, we adopt Depth-First Search (DFS) to establish the traversal order, with only minor differences from standard (pre-order) DFS. Specifically, the initial steps of a typical DFS involve starting from the root node and reaching the first leaf node, marking all intermediate nodes as visited (this can also happen for subtrees). However, our verification starts from the leaf nodes, and a node is marked as visited only after it has been verified. In other words, the verification order is conceptually post-order DFS.

# F   Sequence-level RRSw

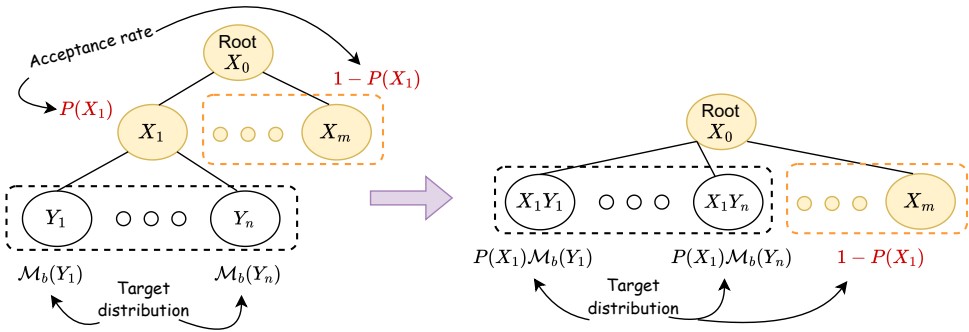

Figure 4: The sequence-level RRSw for two-layers decoding tree.

RRSw is a lossless probability modification method, which recursively redistributes the residual probability to other candidates after rejections, and the probabilities only "flow" within the same layer of a tree. Traversal Verification can be regarded as a sequence-level RRSw. As shown in Figure 4, we first transform the original decoding tree on the left into the right one, and then utilize the classic RRSw algorithm to derive the correct probability transition formulas.

## G  Limitations

Despite Traversal Verification significantly enhances the performance of existing speculative decoding frameworks, there are still some limitations. Firstly, our methodology is fundamentally applied to stochastic decoding scenarios (requiring temperature > 0). In greedy decoding, where the temperature parameter is set to zero, the absence of sampling mechanisms renders all verification approaches functionally equivalent, thereby eliminating any potential performance gains from Traversal Verification. Secondly, the traversal of all tree nodes introduces additional computational overhead during the verification phase. This characteristic may compromise practical throughput in particular environments. However, this issue could be mitigated through optimized implementation, such as discarding the sub-sequences with extremely low probabilities to avoid redundant computational overheads.

## H  Broader Impacts

This paper proposes Traversal Verification, a novel speculative decoding algorithm. Traversal Verification enhances the inference speed of Large Language Models (LLMs), thereby facilitating the deployment on resource-constrained devices such as personal computers, mobile phones, and various edge devices. LLMs themselves may be applied to a wide range of scenarios, potentially leading to various positive or negative societal impacts. This work may indirectly contribute to such impacts, but does not directly produce them.

## I  Licenses for Existing Assets

We summarize the assets and available resources related to this paper in Table 7.

Table 7: Licenses of assets.

| | | |
|---|---|---|
| Models | Llama3.1-8B-Instruct[3] | llama3.1 license |
| | Llama3.2-1B-Instruct[4] | llama3.2 license |
| | Llama2-7B[5] | llama2 license |
| | Llama-68M[6] | apache-2.0 |
| | Vicuna-7B-v1.3[7] | Non-commercial license |
| | EAGLE-Vicuna-7B-v1.3[8] | apache-2.0 |
| Datasets & Codes | Spec-Bench[9] | apache-2.0 |
| | EAGLE[10] | apache-2.0 |

---

[3]https://huggingface.co/meta-llama/Llama-3.1-8B-Instruct

[4]https://huggingface.co/meta-llama/Llama-3.2-1B-Instruct

[5]https://huggingface.co/meta-llama/Llama-2-7b-hf

[6]https://huggingface.co/JackFram/llama-68m

[7]https://huggingface.co/lmsys/vicuna-7b-v1.3

[8]https://huggingface.co/yuhuili/EAGLE-Vicuna-7B-v1.3

[9]https://github.com/hemingkx/Spec-Bench

[10]https://github.com/SafeAILab/EAGLE

