# OpenReview forum: "Traversal Verification for Speculative Tree Decoding"
_NeurIPS.cc/2025/Conference — NeurIPS 2025 poster_

### Official Review · Reviewer_VunK · 2025-06-21

**Clarity:** 3
**Significance:** 3
**Originality:** 3
**Rating:** 4
**Confidence:** 4

**Summary:**

The paper introduces a novel verification method for tree-based speculative decoding. Unlike the conventional verification process, which starts from the root of the tree, the proposed method works backwards from the leaf nodes, thereby boosting the verification success rate with the same lossless guarantee. Theoretically, the paper proved that the proposed method is a lossless verification algorithm. In a single-chain speculation setup, the approach is proven to be optimal in terms of the expected acceptance length. Empirically, the paper demonstrates the effectiveness of their method on multiple models and benchmarks, where the technique can be easily applied on top of existing speculative decoding algorithms.

**Questions:**

1. How do you decide the traversal ordering? I would assume that the traversal ordering can affect your verification success rate. For instance, starting with a path that has a lower KL divergence between the base and draft models might give you a slightly higher verification success rate.

2. Theoretically, the authors prove the optimality under a single chain, which is important. It would be more interesting if the authors could prove that the proposed method is strictly better (not necessarily optimality) than the conventional verification methods that start from the roots. If this is not provable, even adding some assumptions or identifying some scenarios where the proposed method is strictly better than the conventional ones can provide the work with a more solid foundation.

These are general questions that can be open-ended, so it is fine if authors do not provide complete answers. The work at the current stage is already pretty solid and interesting, so I would give a positive rating.

**Ethical Concerns:**

["NO or VERY MINOR ethics concerns only"]

**Final Justification:**

The proposed method is theoretically sound and can be directly applied on top of the existing tree-based verification process to bring additional speed-ups. Thus, I would give a positive rating for the paper.

**Limitations:**

yes

**Quality:**

3

**Strengths And Weaknesses:**

Strengths:

- The presentation is clear and self-contained with illustrative toy examples.
- The problem itself is important and practical, as most of the best-performing speculative decoding methods use a tree-based speculation structure.
- The theoretical guarantee and solid empirical evaluations are a plus.
- The proposed method can be easily adopted to any speculative decoding framework.

Weaknesses:

- The improvements from the evaluation results are marginal.

---

> ### Author Rebuttal · Authors · 2025-07-27
>
> We sincerely appreciate the your insightful review and are pleased to address the raised concerns.
>
> **Q1: How do you decide the traversal ordering? I would assume that the traversal ordering can affect your verification success rate. For instance, starting with a path that has a lower KL divergence between the base and draft models might give you a slightly higher verification success rate.**
>
> We did not intentionally design a specific traversal order. In fact, after the tree structure was determined, we adopted a search method similar to Depth-First Search (DFS) to establish the traversal order, with only minor differences from standard DFS. Specifically, the initial steps of a typical DFS involve starting from the root node and reaching the first leaf node, marking all intermediate nodes as visited (this can also happen for subtrees). We, however, skipped this "root-to-first-leaf" process because our verification starts from the leaf nodes, and a node is marked as visited only after it has been verified. Aside from this, our traversal logic is almost identical to DFS.
>
> Although we did not deliberately adjust the traversal order, this typically doesn't cause any problems. Most existing tree structures are unbalanced, such as static trees designed manually (e.g., the EAGLE sparse tree) or dynamically adjusted tree structures (e.g., Sequoia). These tree structures tend to allocate more and deeper nodes on the left side of the tree, which is intuitive. This is because the original probabilities of draft models are more aligned with the original probabilities of target models, leading to a significantly higher acceptance rate for nodes on the left. Probabilities after RRSw, however, are harder to control. Indeed, experiments in SpecHub (Figure 9) also demonstrate that for RRS/RRSw-based methods, the overall trend of candidate acceptance rate decreases from left to right.
>
> Therefore, under most current tree structures, our verification order is highly efficient. This is because we indeed start from the leftmost leaf node and traverse in a manner similar to DFS. This effectively leverages two advantages of existing tree structures: 1) "deeper on the left" and 2) "higher acceptance rate on the left." In some unusual cases, such as trees intentionally designed to be shallow on the left and deep on the right, using DFS might lead to a decrease in acceptance length, but this is due to the inherent unreasonable nature of the tree structure itself.
>
> Furthermore, we suspect you might be wondering if it's possible to manually adjust the verification order, for example, from an intermediate leaf and intentionally design a traversal order, to achieve a higher acceptance length. However, this is quite difficult because the depth-first traversal order matches the sampling strategy. Manually adjusting the verification order could conflict with the sampling strategy and might lead to a loss of lossless property.
>
> ---
>
> **Q2: Theoretically, the authors prove the optimality under a single chain, which is important. It would be more interesting if the authors could prove that the proposed method is strictly better (not necessarily optimality) than the conventional verification methods that start from the roots. If this is not provable, even adding some assumptions or identifying some scenarios where the proposed method is strictly better than the conventional ones can provide the work with a more solid foundation.**
>
> We've actually considered this problem in great detail both during the writing of this manuscript and after receiving your review. It is, indeed a very charming property which are pursuing for. It's rather easy to demonstrate cases of a single chain, however, for multi-candidate scenarios, even when we envisioned specific scenarios (e.g., the tree's depth strictly non-increasing from left to right), we still couldn't find a good breakthrough to prove it. Therefore, we can only offer some of our current thoughts for discussion.
>
> In our view, there are two primary factors that make demonstrating optimality or supriority extremely challenging.
>
> First, RRSw itself is sub-optimal (although it's one of the best multi-candidate methods and widely adopted by far). After rejecting a token using RRSw, both the remaining $M_b$ and $M_s$ must be adjusted. This adjustment introduces some sub-optimal probability transport, which leads to a non-optimal acceptance rate. So compared with single-chain scenarios, we need to additionally measure "how much acceptance rate is lost" in the RRSw process. Second, the acceptance rate of speculative sampling depends on the overlap between $M_b$ and $M_s$ (which is closely related to KL divergence). However, **the overlaps between the adjusted $M_b'$ and $M_s'$ after rejecting a token are different in token-level Verification and Traversal Verification.** Although Traversal Verification can still benefit from the sequence-level probability advantage, the different distributions after rejection make it quite difficult to prove Traversal Verification's superiority over token-level verification. In other words, compared to "deterministic" single-chain cases, the new probability distribution after rejecting a token using RRSw is "uncontrollable".
>
> So what about considering a different approach? For example, if we temporarily set aside the RRSw algorithm and instead assume there's some optimal multi-candidate method, then combining Traversal Verification with it seems promising. However, finding such an algorithm is actually very tricky, and we aren't even sure if an "optimal multi-candidate algorithm that can be practically employed" truly exists. Therefore, we apologize that we temporally have no idea for this issue, even though we made many attempts so far.

---

> > ### Comment · Reviewer_VunK · 2025-08-05
> >
> > Thanks for the response. I have no further questions. I will keep my score.

---

### Official Review · Reviewer_VhHZ · 2025-07-03

**Clarity:** 3
**Significance:** 2
**Originality:** 3
**Rating:** 4
**Confidence:** 4

**Summary:**

This paper introduces Traversal Verification, a speculative decoding algorithm that accelerates large language model generation by addressing limitations of token-level verification. It claims to guarantee lossless inference and achieves optimality in certain cases. Empirical results show consistent improvements in acceptance length and throughput over existing methods, particularly for larger decoding trees.

**Questions:**

Please address the concerns in Weaknesses.

**Ethical Concerns:**

["NO or VERY MINOR ethics concerns only"]

**Final Justification:**

I appreciate the authors' further clarifications and newly added empirical results, which addressed my major concerns and lead to an updated evaluation. The added evaluation on generation quality clearly enhances the paper. It's also good to know the authors' plan to revise the writing for better clarity.

**Quality:**

2

**Strengths And Weaknesses:**

### Strengths
- The introduced Traversal Verification is a new verification method for speculative decoding with leaf-to-root traversal and sequence-level acceptance.
- The paper provides rigorous proofs for lossless guarantee and optimality in specific scenarios.
- Empirical experiments demonstrates consistent (though trivial) improvements in acceptance length and throughput across various tasks.

### Weaknesses
- The description of traversal verification in Line 45 is problematic. The statement "If the node is accepted, the entire sequence from the current node to the root is accepted" implies a lack of control over the correctness of parent nodes. For instance, in Figure 1, if $X_3$ is accepted, blindly accepting $X_1$ is not guaranteed to be correct.

- The description of tree-based decoding in Line 41 is also problematic, reflecting a misinterpretation of tree-based verification. In standard tree-based verification (e.g., SpecInfer/Medusa/EAGLE), the model verifies multiple sequences (branches from the root to each leaf) **in parallel**, not token by token. It is unlikely that the proposed token-based traversal verification would outperform tree-based verification methods.

- Another major concern is that the experiments only compare with token-level speculative decoding, without comparison to standard tree-based speculative decoding methods (e.g., Medusa, EAGLE) and lack of the autoregressive decoding as a baseline. As a result, the empirical significance is questionable, especially given the trivial speedups shown in Table 1 (typically ~1 token/s faster in terms of throughput).


- The claim in Line 210 that "Traversal Verification is a lossless speculative decoding method" lacks empirical support. To justify this claim, empirical evidence should be provided to confirm the output guarantee consistency with standard autoregressive decoding.

---

> ### Author Rebuttal · Authors · 2025-07-25
>
> Thank you for your detailed review.
>
> **We have carefully read your questions and found that you may have some misunderstandings on our method, please allow us to begin with an example to thoroughly explain what we are doing and why we design Traversal Verification. This should help clarify your misunderstanding and illustrate our motivation.**
>
> Let's first discuss about the case of a single chain. Suppose we sample a sequence of length 2, where
>
> * Draft model probability ($M_s$) = 0.9 (for the first token) and 0.1 (for the second token)
> * Target model probability ($M_b$) = 0.1 (for the first token) and 0.9 (for the second token)
>
> For conventional speculative decoding, we first consider whether the first token is acceptable. If it is, we then move on to the second token (Of course you can do this in parallel, but it doesn't change the essence). This is why we refer to this as "token-level verification." So obviously, the acceptance rate for this sequence would be $1/9 \times 1 = 0.11$. However, if we consider the probability of the **entire sequence**, the acceptance probability is $(0.9 \times 0.1) / (0.1 \times 0.9) = 1.0$. This means the entire sequence could be accepted directly.
>
> Because **token-level verification only considers the acceptance or rejection of individual tokens, it loses the optimality at the sequence level.** This is also a problem inherit in token-level verification: the cumulated acceptance rate is monotonically non-increasing (because the acceptance probability of a child node is always no more than 1). This clearly overlooks situations of probability "crossover". As shown in our example above, since $M_b > M_s$ at the second position, a higher acceptance rate for the second position compared to the first is permissible when considering the sequence-level probability distribution – **meaning, even if the first token is rejected, its child and the total sequence still have a probability to be accepted.**
>
> By this example, we assume that you have understood that considering the "overall probability" of a sequence, instead of merely the "per-token" probability, may lead to a higher acceptance length, and **the only problem is "how to use the sequence-level probability while maintaining losslessness". In fact, it's realizable, as both in this paper and in Block Verification (ICLR 25) found ways to solve it.**
>
> Next, let's discuss the multi-candidate scenario. When multiple candidates exist at each depth of a chain, it forms a decoding tree. Current methods typically employ Recursive Rejection Sampling (RRS, represented by SpecInfer) and **RRS without replacement (RRSw, employed by EAGLE, Sequoia, MCSD, etc.)**. These multi-candidate approaches are also known as "multi-round token verification," which is an extension of the single-candidate case. Essentially, it remains token-level verification, simply involving multiple rounds of rejection-resampling.
>
> **Think of it like building a house: token-level verification is the foundation, while specific decoding method, such as EAGLE, SpecInfer, are the houses constructed on top of this foundation. In fact, nearly all existing methods (except for Asps and Block Verification, but they only discuss the single-chain cases) rely on token-level verification**.
>
> **What we are doing is to "extend single-chain sequence-level verification to the tree cases", and all tree decoding methods using token-level verification can benefit from this new "foundation"**. This is quite challenging because you need to carefully modify the probablity after rejecting a token to ensure losslessness.
>
> **In this paper, we leverage a leaf-to-root traversal mechanism to make full use of sequence-level probability, and design a quite delicate probability adjustment strategy to maintain losslessness.**
>
> Thus, Traversal Verification is actually on a more basic level than specific methods like EAGLE, Medusa, and SpecInfer. **Our aim is to addresses the suboptimality of multi-round token verification methods in tree decoding, thereby enhance all the token-level based methods including EAGLE, rather than propose a specific method parallel to EAGLE.**
>
> Now let's look at the questions you raised.
>
> ---
>
> **Q3. The experiments only compare with token-level speculative decoding, without comparison to standard tree-based speculative decoding methods and lack of the autoregressive decoding as a baseline.**
>
>  *You may now be able to understand that what we refer to as "token-level speculative decoding" in our paper refers to all token-level verification methods including EAGLE and SpecInfer.*
>
> Regarding Medusa, its uses either token-level verification (lossless) or their proposed "typical sampling" (lossy, but faster) as the verification strategy, while token-level verification based methods such as EAGLE and SpecInfer are completely lossless. Therefore, we do not consider Medusa a proper reference for lossless speculative decoding.
>
> Regarding to auto-regressive baseline, we have provided the generation speed for Llama3 and Llama2 in line 222-223, maybe you have missed them.
>
> ---
>
> **Q1. The statement "If the node is accepted, the entire sequence from the current node to the root is accepted" implies a lack of control over the correctness of parent nodes. For instance, in Figure 1, if $X_1$ is accepted, blindly accepting $X_3$ is not guaranteed to be correct.**
>
> *As we've discussed, this is exactly our core contribution. We are not "blindly" accepting a token, but with careful probability modification to ensure losslessness.*
>
> With token-level verification, you must accept $X_1$ before you can accept $X_3$. This is the implication you've made in your question. However, our paper utilizes sequence-level probability and addresses this limitation:even if $X_1$ is rejected, the sequence $X_1X_3$ can still be accepted.
>
> **The ultimate goal of speculative decoding is to maintain the same probability as the original model**. $X_1$ being rejected doesn't mean it's "incorrect"; the root cuase is that, under token-level verification, you **must** reject it to maintain consistency with the original distribution. Conversely, if you consider sequence-level probabilities, **the presence of subsequent tokens can, in some cases, allow you to keep $X_1$ while maintaining the sequence's probability consistent with the original**.
>
> ---
>
> **Q2. The description of tree-based decoding in Line 41 is also problematic, reflecting a misinterpretation of tree-based verification. In standard tree-based verification (e.g., SpecInfer/Medusa/EAGLE), the model verifies multiple sequences in parallel, not token by token.**
>
> *We are confident that our understanding of tree-based decoding is accurate. Let's discuss the verification process now.*
>
> The verification process, in fact, consists of two distinct steps:
>
> 1.  **Step 1: Feeding drafted tokens into the target model.** The drafted tokens are fed into the target model in **parallel** to obtain the target distribution. Almost all methods follows this mechanism, such as SpecInfer, Medusa and EAGLE, and it is the same for our method.
> 2.  **Step 2: Acceptance decision.** Based on the target and draft model distributions, a verification is performed. In some cases (e.g., chain, and maybe typical sampling in Medusa) you can perform this in parallel. However, for RRS/RRSw (such as EAGLE), parallel verification is difficult because after a token is rejected, the residual distribution (i.e., $[M_b - M_s]^+$) needs to be calculated as the new target distribution, and you need to recursively obtain the residual distribution after each rejection. You cannot verify all sequences in parallel unless you pre-calculate those residual probabilities, and this is unnecessary and not economic.
>
> **Regarding your mention of EAGLE, we recommend that you examine its source code to confirm the acceptance logic**. You can check the “evaluate_posterior” function in eagle/model/utils.py (either in the main or v1 branch) within the official EAGLE repository. **You will find that it determines acceptance token-by-token**.
>
> **Furthermore, we suggest you refer to Algorithm 2 on page 5 of the SpecInfer paper and Algorithm 1 on page 13 of the EAGLE paper**. Both clearly describe the execution logic of SpecInfer or EAGLE's verification: they verify token by token in the same layer using token-level probabilities, and adjust the remaining probabilities after rejecting a candidate.
>
> **Therefore, we do not believe there is any issue with the statement in Line 41**.
>
> Beyond this, **as we have stated in Line 211 of this paper, the baseline token-level verification method we compared against in our experiment is indeed EAGLE's tree verification implementation.**
>
> Consequently, your statement that "It is unlikely that the proposed token-based traversal verification would outperform tree-based verification methods" is incorrect. Our experiments clearly demonstrate an advantage over EAGLE.
>
> ---
>
> **Q4. The claim in Line 210 that "Traversal Verification is a lossless speculative decoding method" lacks empirical support. To justify this claim, empirical evidence should be provided to confirm the output guarantee consistency with standard autoregressive decoding.**
>
> *As you highlighted in the Strengths, we have already provided a complete, mathematically rigorous proof of the losslessness of our method.*
>
> **Most existing mathematically lossless methods, such as EAGLE, Sequoia, SpecInfer, Block Verification, even the foundational paper in this field, "Fast Inference from Transformers via Speculative Decoding", does not empirically evaluate the losslessness.** Conversely, attempting to prove losslessness through experiments would be unconvincing. Factors like sampling variability and hardware fluctuations can introduce randomness into experimental results. Consequently, experiments alone cannot definitively prove or disprove whether a method is lossless.
>
> We hope our explanations could address your concerns.

---

> ### Comment · Reviewer_mJgv · 2025-08-04
>
> Hi, thank you for taking the time to review this submission and for sharing your concerns about correctness.
>
> > For instance, in Figure 1, if $X_3$ is accepted, blindly accepting $X_1$ is not guaranteed to be correct.
>
> I'm not sure I fully understand this one. In prior work like "BLOCK VERIFICATION ACCELERATES SPECULATIVE DECODING", accepting a later token also implies that all tokens from $X_1$ to $X_\tau$ are accepted, so that’s a standard in prior work.
>
> Could you please clarify how exactly the soundness is broken? Looking forward to discussing this further during the reviewing process. Thanks again for your time and insights!

---

> ### Comment · Reviewer_VhHZ · 2025-08-04
>
> Thank the authors for the detailed clarifications. I still have two concerns:
>
> First, the proposed verification method is limited to sampling-based decoding (temp > 0) and will lead to inconsistent decoding results with standard decoding under greedy decoding (temp = 0). Under greedy decoding, there is only one deterministic token at each position. The following claim is clearly **incorrect**:
> > even if $X_1$ is rejected, the sequence $X_1X_3$ can still be accepted.
>
> When $X_1$ is rejected, meaning that $X_1$ is not aligned with the ground-truth $X_1^*$, accepting the sequence $X_1X_3$ will lead to inconsistent output with the standard decoding. I'd suggest the authors revise this claim with clearer contexts in the paper.
>
> Second, given that the method only works under non-greedy decoding setup, meaning that the model will generate different outputs even for the same input. **It's crucial to evaluate the generation quality, as also pointed out by reviewers QRvg.**
>
> Without such an empirical validation, it is unclear whether the model will compromise the generation quality for speedups. I don't mean to be nitpicking, but just because some previous work (e.g., EAGLE) didn't evaluate generation quality doesn't mean it's a standard practice to follow. Many impactful works (e.g., Medusa) have reported both generation speed and generation quality.
>
> That said, considering many of them are token-based verification (e.g, EAGLE), it's understandable not to report the generation quality again since such token-based verification has proved in many prior works. **However, the proposed traversal verification is sequence-based, so it cannot assume the same output generation quality guarantee as preserved in token-based verification.** Thus, it's necessary to provide empirical validation to justify the lossless claim.
>
> Thanks Reviewer mJgv for joining the discussion. Please refer to point 1 in the above response for more details. I'm open to further discussion and willing to reconsider my evaluation if new points are raised.

---

> ### Author Response · Authors · 2025-08-05
> **Response to Reviewer VhHZ (1/2): General Responses**
>
> Thank you for your comments. We are pleased to address your concerns.
>
> **Q1. Traversal Verification under Greedy Decoding (Temperature=0).**
>
> During the writing and rebuttal process for this paper, some of our phrasing may have been imprecise for the sake of a more intuitive explanation. We apologize if they caused any misunderstandings. Regarding the sentence you quoted, our intention was to express that **under certain circumstances**, even after rejecting $X_1$, there the sequence $X_1X_3$ could still be accepted. **Essentially, this "acceptance" must be subject to the acceptance probability determined in our Traversal Verification algorithm.**
>
> We state our conclusion here and will provide a concise proof at the end of our response: **When temp = 0, only the top-1 token should be accepted. In this case, Traversal Verification *is equivalent to* standard token-level verification under Greedy Decoding. Traversal Verification remains lossless regardless of whether the temperature is zero or not.**
>
> Referring to the initialization and modification formulas in our Traversal Verification algorithm, under greedy decoding, the acceptance rate of the top-1 token is 1, while the acceptance rates of other tokens are 0. Moreover, if any node in a sequence has an acceptance rate of 0, the subsequent nodes' sequence acceptance rate (i.e., the cumulative probability) will be 0. Although Traversal Verification still executes verification checks on these zero-acceptance-rate sequences, this does not affect the final acceptance result: we never accept non-top-1 tokens or their subsequent child tokens (under temp = 0).  *In other words, under greedy decoding, if the acceptance rate of $X_1$ is 0, the acceptance rate of $X_1X_3$ is also 0, which is in alignment with token-level verification.*
>
> **We will carefully consider the wording in the final version of the paper to ensure its rigor**. Thank you for your valuable suggestions!
>
> **Q2. Generation Quality**
>
> We understand your concerns about generation quality. While we have mathematically proven the lossless nature of Traversal Verification (meaning the output's probability distribution is identical to the original), we acknowledge that Traversal Verification is indeed different from most previous token-level verification methods, and we believe your concerns are entirely reasonable.
>
> Based on your suggestions, **we have followed the method used in Medusa for measuring generation quality: we use the MT-bench dataset and employed a state-of-the-art LLM as a judge to evaluate the quality of the generated answers.**
>
> We use Gemini-2.5-Flash as the judge model to assess the quality of the MT-bench responses. Please forgive us for not using the GPT series as the judge model. (This is due to the significant cost of running the judge of quality, and we are currently only able to expense Gemini bills due to our reimbursement policy)
>
> Nevertheless, using Gemini as a judge is definitely reliable. Gemini-2.5-Flash is one of the representative state-of-the-art models, with performance on text generation tasks that is nearly identical to GPT-4.1-2025-04-14 (see the LMArena Leaderboard, where both models scored 1408), and it significantly outperforms the GPT-4 used in the Medusa and MT-bench last year (which only scored a bit more than 1300).
>
> Below are the results of our generation quality evaluation on MT-bench with Llama3.1-8B-Instruct (using the same 10-point scale as Medusa, higher score is better). For all experiments, we ran them three times and report the average.
>
> | | Verification Strategy | Quality |
> | :--- | :--- | :--- |
> | auto-regressive | N/A | 6.72 |
> | chain | token-level | 6.78 |
> | | traversal | 6.77 |
> | eagle sparse tree | token-level | 6.76 |
> | | traversal | 6.79 |
> | binary tree | token-level | 6.69 |
> | | traversal | 6.74 |
>
> **As you can see, Traversal Verification, EAGLE (token-level), and naive generation all achieved scores at the same level (it's expected to have some fluctuations due to the randomness of sampling). Therefore, we can conclude that Traversal Verification maintains generation quality.**
>
> We hope that our explanation and experimental results can address your concerns regarding temperature and generation quality.
>
> Thank you again for your valuable feedback, and we are willing to discuss if you have further concerns.

---

> ### Author Response · Authors · 2025-08-05
> **Response to Reviewer VhHZ (2/2): Proof of Equivalence under Temp=0**
>
> Now, we provide the proof demonstrating equivalence under temp=0. Fundamentally, standard token-level verification can also be viewed as a sequence-level method. The essential distinction between it and Traversal Verification lies in their node acceptance probability formulas.
>
> Let $X$ denote the current node, $p_i$ be its sequence-based acceptance probability, and $p_{i-1}$ be its parent node's sequence-based acceptance probability (with $p_0=1$), then
>
> Token-level Verification's formula: $p_i = p_{i-1} \cdot \min \lbrace \frac{M_b(X)}{M_s(X)}, 1\rbrace$,
>
> Traversal Verification's formula: $p_i = \min \lbrace p_{i-1} \cdot \frac{M_b(X)}{M_s(X)}, 1\rbrace$.
>
> **Proof by mathematical induction:**
>
> 1) **Base case (i=0):** $p_0^{tok}=p_0^{tra} \in \lbrace 0, 1 \rbrace$.
>
> 2) **Inductive step:** Assume $p_i^{tok}=p_i^{tra} \in \lbrace 0, 1 \rbrace$ holds for $i=k$, then
> $p_{k+1}^{tok} = 0\~(\text{if } p_k^{tok}=0) \text{ or } p_{k+1}^{tok}=\min \lbrace \frac{M_b(X)}{M_s(X)}, 1\rbrace\~(\text{if } p_k^{tok}=1)$,
> $p_{k+1}^{tra} = 0\~(\text{if } p_k^{tra}=0) \text{ or } p_{k+1}^{tra}=\min \lbrace \frac{M_b(X)}{M_s(X)}, 1\rbrace\~(\text{if } p_k^{tra}=1)$.
>
> Under greedy decoding, we have $\min \lbrace \frac{M_b(X)}{M_s(X)}, 1\rbrace \in \lbrace 0, 1 \rbrace$. Together with the inductive assumption $p_i^{tok}=p_i^{tra} \in \lbrace 0, 1 \rbrace$, it is proved that $p_i^{tok}=p_i^{tra} \in \lbrace 0, 1 \rbrace$ for all $i$.
>
> **Corollary:** When token-level verification assigns a parent node acceptance probability 0, all descendant nodes necessarily have acceptance probability 0. Since traversal verification yields identical acceptance probabilities to token-level verification under greedy decoding, it cannot accept any node rejected by token-level verification.

---

> > ### Comment · Reviewer_VhHZ · 2025-08-05
> >
> > I appreciate the authors' further clarifications and newly added empirical results, which addressed my major concerns and lead to an updated evaluation. The added evaluation on generation quality clearly enhances the paper. It's also good to know the authors' plan to revise the writing for better clarity.

---

### Official Review · Reviewer_QRvg · 2025-07-03

**Clarity:** 3
**Significance:** 3
**Originality:** 3
**Rating:** 4
**Confidence:** 3

**Summary:**

This paper focuses on the sepeclatvie decoding to accelerate LLMs and  tree verification is a common technique in speculative decoding to further improve the acceleration. The authors identify the two potential problems in the current speculative tree decoding, including token-level and root-to-leaf. The authors accordingly propose the Traversal Verification to address the above two problems.

**Questions:**

See above weaknesses and questions.

**Ethical Concerns:**

["NO or VERY MINOR ethics concerns only"]

**Final Justification:**

Based on the rebuttal, my concerns are addressed.  I raise the score to 4.

**Limitations:**

yes

**Quality:**

2

**Strengths And Weaknesses:**

Strengths

1.	The motivation of this paper is solid. There is indeed potential waste in the token tree. If these latent tokens can be effectively utilized, there is a chance for some improvement.

2.	There is a theoretical argument demonstrating that Traversal Verification is lossless. I believe this is crucial for Traversal Verification, as it seems difficult to ensure losslessness if the verification relies solely on accepting leaf nodes.

Weaknesses and questions

1.	Could the authors introduce more baselines for comparison? Currently, only some basic native methods are used as baselines. Since the authors utilize the widely-used SpecBench dataset, there should be many existing baselines whose results can be reproduced for comparison.

2.	The authors should apply the proposed Traversal Verification to other speculative tree decoding methods to demonstrate its generalizability. Since the authors claim this is a plug-and-play method, this should be validated through experiments.

3.	The authors state that “Notably, since Traversal Verification is a lossless speculative decoding method, there is no need to evaluate the generation quality.” I disagree with this. Most previous speculative decoding methods are also lossless, yet they still report generation quality in their experiments.

4.	Regarding the acceleration metrics, I believe the authors follow previous work in reporting the speedup performance. The accept length and token generation per second do not align with the metrics in SpecBench.

---

> ### Author Rebuttal · Authors · 2025-07-25
>
> Thank you for your detailed review.
>
> We understand your concerns regarding the experiment, but we believe there are some things that need clarification. Therefore, before answering your questions, we'd like to describe the motivation and methods of this paper in a simpler and more intuitive way to help you understand, and then we will address your questions.
>
> Let's first discuss about the case of a single chain. Suppose we sample a sequence of length 2, where
>
> * Draft model probability ($M_s$) = 0.9 (for the first token) and 0.1 (for the second token)
> * Target model probability ($M_b$) = 0.1 (for the first token) and 0.9 (for the second token)
>
> For conventional speculative decoding, we first consider whether the first token is acceptable. If it is, we then move on to the second token. This is why we refer to this as "token-level verification." So obviously, the acceptance rate for this sequence would be $1/9 \times 1 = 0.11$. However, if we consider the probability of the **entire sequence**, the acceptance probability is $(0.9 \times 0.1) / (0.1 \times 0.9) = 1.0$. This means the entire sequence could be accepted directly.
>
> Because **token-level verification only considers the acceptance or rejection of individual tokens, it loses the optimality at the sequence level.** This is also a problem inherit in token-level verification: the cumulated acceptance rate is monotonically non-increasing (because the acceptance probability of a child node is always no more than 1). This clearly overlooks situations of probability "crossover". As shown in our example above, since $M_b > M_s$ at the second position, a higher acceptance rate for the second position compared to the first is permissible when considering the sequence-level probability distribution – **meaning, even if the first token is rejected, its child and the total sequence still have a probability to be accepted.**
>
> By this example, we assume that you have understood that considering the "overall probability" of a sequence, instead of merely the "per-token" probability, may lead to a higher acceptance length, and **the only problem is "how to use the sequence-level probability while maintaining losslessness". In fact, it's realizable, as both in this paper and in Block Verification (ICLR 25) found ways to solve it.**
>
> Next, let's discuss the multi-candidate scenario. When multiple candidates exist at each depth of a chain, it forms a decoding tree. Current methods typically employ Recursive Rejection Sampling (RRS, represented by SpecInfer) and Recursive Rejection Sampling without replacement (RRSw, represented by EAGLE, Sequoia, MCSD, etc.). These multi-candidate approaches are also known as "multi-round token verification," which is an extension of the single-candidate case. Essentially, it remains token-level verification, simply involving multiple rounds of rejection-resampling.
>
> **Think of it like building a house: token-level verification is the foundation, while specific decoding method, such as EAGLE, SpecInfer, are the houses constructed on top of this foundation. In fact, nearly all existing methods (except for Asps and Block Verification, but they only discuss the single-chain cases) rely on token-level verification.**
>
> **So, what we are doing is to "extend single-chain sequence-level verification to the tree cases", which is an enhancement of this foundation. It's not to propose a specific method, nor a plug-in for tree decoding methods.** This is quite challenging because you need to carefully modify the residual probability after rejecting a token to ensure losslessness.
>
> **In this paper, we leverage a leaf-to-root traversal mechanism to make full use of sequence-level probability, and design a quite delicate probability adjustment strategy to maintain losslessness.**
>
> We really appreciate that you acknowleged our innovation of tree traversal, but **we want to highlight that our core innovation is to reform the fundamental verification algorithm, and the traversal mechanism serves as a good means to make use of the sequence-level probability.**
>
> Now let's look at the questions you raised.
>
> ---
>
> **Q1. Could the authors introduce more baselines for comparison? Currently, only some basic native methods are used as baselines.** and **Q2. The authors should apply the proposed Traversal Verification to other speculative tree decoding methods to demonstrate its generalizability.**
>
> Now, you may see that our **baseline is not "a specific tree decoding method" but rather "the speculative sampling algorithm itself and its suboptimality under tree decoding conditions."**
>
> We can cite numerous examples that use token-level verifications:
>
> For chain, examples include original Speculative Decoding, REST, and so on.
>
> For tree, examples include EAGLE, SpecInfer, Sequoia, and MCSD, etc.
>
> **All these methods are on top of token-level verification, which serves as the base. What we are doing is not to build a decoration or an auxiliary on specific tree decoding methods, but to reconstruct the basic algorithm of existing token-level speculative decoding. That's why reviewer mJgv highly praised our work and said we make a valuable contribution to speculative decoding research.**
>
> Therefore, your two challenges, concerning generalization and more baselines, are essentially the same question. **In reality, we only have one baseline: the token-level verification foundation**. No matter how the "houses" (specific speculative decoding methods) built on top change, it doesn't affect the validity of the foundation.
>
> We have mathematically proven the following:
>
> 1.  **For chain decoding cases, the expected acceptance length of Traversal Verification is strictly greater than or equal to token-level verification, and it achieves optimal acceptance length**. (It's almost impossible to find or prove the optimality for tree due to uncertainty in multi-candidates situations)
> 2.  **Traversal Verification is lossless.**
>
> Our mathematical proof of optimality provides strong support for the generalization of our method. The only question is how much stronger it is compared to token-level verification, for example, with different draft models, the improvement might vary.
>
> We emphasize again, **Traversal Verification stands alongside vanilla speculative sampling (token-level verification) as a fundamental verification logic, not a decoration or auxiliary for specific methods.** In our experiments, we've already used various tree structures (chain, binary tree, EAGLE tree) and different model series (e.g., Llama3, Llama2) to demonstrate and compare the improvements of our method. You can, of course, use other model as drafters, such as the methods evaluated on Spec-bench using different draft models or different tree structures, but it is not neccesary, as Traversal Verification is on a more basic level than the specific methods, and **our effectiveness comes from the sequence-level probability, and it's generalizability is mathematically orthorgonal to those "houses".**
>
> ---
>
> **Q3. The authors state that “Notably, since Traversal Verification is a lossless speculative decoding method, there is no need to evaluate the generation quality.” I disagree with this. Most previous speculative decoding methods are also lossless, yet they still report generation quality in their experiments.**
>
> Let's start by quoting a passage from the original EAGLE paper, currently one of the most well-known speculative decoding methods: "**Acceleration of EAGLE theoretically guarantees the preservation of the target LLMs’ output distribution. Consequently, evaluating the quality of EAGLE’s generated results is both unnecessary and meaningless.**" (Page 5, Line 12 of the right colomn).
>
> As far as we know, most well-known existing lossless methods such as EAGLE (ICML 24), Sequoia (NeurIPS 24), SpecInfer (ASPLOS 24), Block Verification (ICLR 25), even the foundational paper in this field, "Fast Inference from Transformers via Speculative Decoding," didn't measure generation quality.
>
> You might point out that Medusa series states the quality. However, this is because Medusa proposes "typical acceptance" (which is not a lossless verification strategy), and Medusa-2 even train the backbone's weights (See section 2.2.2 in the Medusa paper), so measuring the generation quality is definitely necessary.
>
> We think your statement that "Most previous speculative decoding methods are also lossless, yet they still report generation quality in their experiments" is questionable, as we have not encountered any well-known published works supporting this claim. We would greatly appreciate it if you could provide some examples.
>
> ---
>
> **Q4. Regarding the acceleration metrics, I believe the authors follow previous work in reporting the speedup performance. The accept length and token generation per second do not align with the metrics in SpecBench.**
>
> We are a little confused that you mentioned "the accept length and token generation per second do not align with the metrics in SpecBench".
>
> First, we are not using the same drafters. When selecting models, we chose the widely adopted and easily accessible Llama2 & Llama3 series as our target and draft models. We select these general, widely used model to strengthen the persuasiveness of our paper, as it focuses on "optimizing the foundation." Spec-Bench covers a lot of NLP tasks, and we use it as a comprehensive dataset to evaluate and compare our method with token-level verification.
>
> Second, the actual performance of different methods is only meaningful when compared under the same hardware conditions.
>
> Since our drafters are different from the specific methods on Spec-Bench, and even our GPUs are different, thus, we are confused about what "not align with" means here, as it's impossible to find a method with the same experimental conditions as ours on Spec-Bench.
>
> We hope our explanations could address your concerns and provide a clearer demonstration of this paper.

---

> ### Author Response · Authors · 2025-08-05
>
> Dear Reviewer QRvg,
>
> We have just provided experimental results regarding the generation quality of Traversal Verification in our response to Reviewer VhHZ. We kindly invite you to examine our results, as they may help address your concerns. Thank you!

---

> ### Author Response · Authors · 2025-08-06
>
> Hi Reviewer QRvg,
>
> We have conducted additional experiments to improve our persuasiveness, in hope of further addressing your concerns!
>
> **Concern 1: Discussion on direct comparison with reported performance on SpecBench leaderboard**
>
> We previously mentioned that even with the same draft and target models, the final speedup and acceptance length can vary across different hardware. To address the concerns you might have, **we conducted an experiment using Vicuna-7B-v1.3 as the target model and EAGLE-vicuna as the draft model, consistent with the official SpecBench leaderboard**, on an A6000 GPU. Our results show an acceptance length and speedup of 3.56 and 2.20x, respectively. The results are roughly comparable to the results reported on SpecBench for the 3090 GPU (3.57/2.03x) and A100 GPU(3.58/2.10x), while some differences still remain.
>
> Also, a direct comparison would be even less meaningful if both the target and draft models are different. **For example, our paper uses the widely used Llama3 series, which has not been tested on the official SpecBench leaderboard so far.**
>
> We also want to point out that the acceptance length and speedup reported on the SpecBench leaderboard are based on Temperature=0 (greedy decoding). Traversal Verification, which uses sequence-level probability, is primarily designed for non-greedy decoding. (As we proved in our response to reviewer VhHZ, Traversal Verification is equivalent to token-level verification when Temperature=0.) **Due to the inherent uncertainty in sampling during non-greedy decoding, the acceptance length and speedup will be generally slightly lower than those in greedy decoding.** You can find corresponding experimental results on this matter in the EAGLE paper.
>
> Based on these factors, it’s not proper to directly compare the speedup or acceptance length with the reported results on distinct models, different devices, or varying temperatures. We used SpecBench because it covers many NLP tasks, and our primary goal was to test the effectiveness of our method, not to compare it with specific techniques or to top a leaderboard.
>
> **Concern 2: Experiments on other draft model or baseline**
>
> As we mentioned in our rebuttal, our baseline is essentially the speculative decoding algorithm. **The effectiveness of our method is mathematically guaranteed by using sequence-level probability, making it independent of any specific methods**.
>
> **Nevertheless, we've included our experimental results on Vicuna-7B-v1.3 (consistent with SpecBench official leaderboard), using EAGLE-vicuna as the draft model, with Temperature=1.**
>
> The default tree structure is EAGLE sparse tree, and the default verification of EAGLE is token-level verification (RRSw).
>
> | Method | Acceptance Length | Speedup |
> | --- | --- | --- |
> | EAGLE (default) | 3.18 | 2.00x |
> | **EAGLE (traversal)** | **3.26** | **2.04x** |
>
> For other tree structures, you can also enjoy the improvement of Traversal Verification.
>
> **Chain:**
>
> | Method | Acceptance Length | Speedup |
> | --- | --- | --- |
> | EAGLE (default) | 2.57 | 1.77x |
> | **EAGLE (traversal)** | **2.63** | **1.81x** |
>
> **Binary tree:**
>
> | Method | Acceptance Length | Speedup |
> | --- | --- | --- |
> | EAGLE (default) | 3.11 | 1.87x |
> | **EAGLE (traversal)** | **3.22** | **1.92x** |
>
> As you can see, our method is also effective with the EAGLE draft model, providing a stable and consistent improvement.
>
> **Concern 3: Generation Quality**
>
> As we mentioned in our previous response, we have already addressed the concerns about generation quality by including relevant experiments in our reply to Reviewer VhHZ. We have copied the result here for your convenience.
>
> The generation quality is measured on MT-bench dataset using LLM as a judge, following the previous work such as Medusa. Our metric is also aligned with Medusa, using a 10-point scale, and a higher score represents a better quality.
>
> |  | Verification Strategy | Quality |
> | --- | --- | --- |
> | auto-regressive | N/A | 6.72 |
> | chain | token-level | 6.78 |
> |  | traversal | 6.77 |
> | eagle sparse tree | token-level | 6.76 |
> |  | traversal | 6.79 |
> | binary tree | token-level | 6.69 |
> |  | traversal | 6.74 |
>
> The results show that Traversal Verification maintains the same generation quality as both naive generation and EAGLE (it's reasonable to have some fluctuations due to sampling), which serves as strong evidence for its **lossless** property.
>
> We hope this response clarifies your concerns. Please do not hesitate to reach out if you have any further questions!

---

> > ### Comment · Reviewer_QRvg · 2025-08-06
> >
> > Sorry for the late reply. I am convinced by the second reply with experiments to address my concern. I will raise the score.
> >
> > Additionally, I have a few comments:
> >
> > 1) The first reply suggests that "my method is theoretically guaranteed and does not require experiments". Is this really valid, especially in the AI field? I believe it is not. For example, we can observe that the proposed method achieves different speed-ups with different draft models. If you do not include more draft models, how can you verify that the proposed method generalizes well to different scenarios?
> >
> > 2) Why do we need a benchmark? Besides providing data for everyone to test, the more important reason is that the baseline results are reported by others. You can add new draft models, but why not also include the old draft models to align with the benchmark and increase the credibility of your method?
> >
> > This is why I asked these questions earlier. I need the experiments to help me make a judgment, rather than being told that these experiments are meaningless.

---

> > > ### Author Response · Authors · 2025-08-07
> > >
> > > Thank you so much for your reply! We completely agree with your feedback.
> > >
> > > In our initial response, we focused too much on the mathematical foundations and theoretical explanations while neglecting the importance of experiments. We reflected on this point during the discussion phase and realized the shortcomings of our initial response, and then decided to provide empirical results to strengthen our arguments.
> > >
> > > We are very fortunate that the supplementary experiments have been acknowledged by you. We will add these results to the final version of our paper.
> > >
> > > Thank you again for your valuable suggestions. We will take them to heart.

---

### Official Review · Reviewer_mJgv · 2025-07-07

**Clarity:** 3
**Significance:** 3
**Originality:** 3
**Rating:** 5
**Confidence:** 3

**Summary:**

This paper considers the speculative decoding problem with multiple candidates and multiple steps. The authors propose a new technique that achieves higher sampling efficiency than existing work. The effectiveness of the proposed method is validated through extensive experiments.

**Questions:**

No

**Ethical Concerns:**

["NO or VERY MINOR ethics concerns only"]

**Final Justification:**

I believe this paper makes a valuable contribution to speculative decoding research. I will maintain my original score and recommend its acceptance.

**Limitations:**

While not a weakness, there is still room for improvement in this method. RRSw is not optimal for multiple candidates, though finding the optimal solution for multiple candidates is indeed very challenging. Nevertheless, this work represents sufficient progress in the field and makes a valuable contribution to speculative decoding research.

**Quality:**

3

**Strengths And Weaknesses:**

The method cleverly combines optimal single-chain verification with RRSw (Recursive Rejection Sampling without replacement), to handle tree-structured candidates with best efficiency so far.

While I haven't checked all proofs in detail, the construction and theoretical framework look sound to me. The method recovers the optimal single-chain verification as a special case. When rejection occurs, it properly modifies the distribution on the tree to maintain the optimality property.

The experimental validation is comprehensive.

Just a very minor weakness: Since the method combines optimal single-chain verification with RRSw, the baseline comparisons could also include optimal single-chain verification in addition to RRSw. Although using single chains without trees would likely perform poorly and may not be necessary to show, it could provide additional context.

---

> ### Author Rebuttal · Authors · 2025-07-25
>
> We deeply appreciate your insightful review and are pleased to address your comments regarding our work.
>
> As you have pointed out, Traversal Verification is essentially an extention of optimal single-chain verification to multi-candidate scenarios (tree decoding). By delicate probability modification, which is a specially-designed RRSw, we maintain lossless acceleration.
>
> In fact, we had concerned that the paper might be overly mathematical and challenging for readers unfamiliar with the field, leading us to incorporate reader-friendly examples and detailed explanations to enhance clarity. However, your review demonstrates a deep understanding of this field, and directly extracts our core idea, and we would like to show our respect for your expertise in this field.
>
> Below are our responses to your specific comments:
>
> 1.  **There is still room for improvement in this method. RRSw is not optimal for multiple candidates, though finding the optimal solution for multiple candidates is indeed very challenging.**
>
>     We completely agree with your assessment. This observation aligns perfectly with our ongoing research. In multi-candidate scenarios, RRSw is not optimal and can indeed incur some losses in acceptance rate when trasporting probability. Existing works, such as SpecHub and Greedy selection, attempt to mitigate this issue, and we find this to be a very interesting direction for future exploration.
>
> 2.  **Since the method combines optimal single-chain verification with RRSw, The baseline comparisons could also include optimal single-chain verification in addition to RRSw.**
>
>     Thank you for your valuable suggestion. We agree that including this comparison would undoubtedly enrich the paper. Although we have mathematically demonstrated that Traversal Verification is equivalent to optimal single-chain verification (i.e., Block Verification) in the single-chain case, minor discrepancies in experimental results might arise due to factors like sampling randomness. We will consider adding these experiments to the final version.
>
> **Thank you once again, for your expertise in speculative decoding and the positive feedback on our manuscript.**

---

### Comment · Area_Chair_fNid · 2025-08-06

Dear Reviewers,

This is a quick reminder that we are now in the post-rebuttal discussion phase.

Please take the time to read the author rebuttal and engage in discussion with the other reviewers. Your input is crucial for us to make a final, informed decision as the deadline is approaching.

Thank you for your timely participation.

Best regards,

Area Chair

---

### Author Response · Authors · 2025-08-07
**Summary of Discussion**

Thank you very much to all the reviewers for reviewing our paper and participating in the discussion! We have benefited greatly from your feedback.

We are pleased that all reviewers' concerns have been addressed. Here we provide a brief summary of our paper and the rebuttal & discussion for ease of reading.

This paper proposes Traversal Verification, a novel speculative decoding algorithm. Almost all existing speculative decoding algorithms use token-level probabilities to decide whether to accept a drafted token, which sacrifices the optimality of the sequence-level acceptance length. While a few works (such as ASpS and Block Verification) have considered sequence-level probabilities, they have focused on chain decoding and have not addressed the case of tree decoding (i.e., the multi-candidate scenario).

Our work fills this gap in the research field. We rethink the sub-optimality of token-level verification in tree decoding and design a traversal mechanism to effectively leverage sequence-level probabilities. By meticulously controlling the acceptance rate and residual probability, we achieve a higher acceptance length and greater speedup while maintaining losslessness. We provide a rigorous proof of Traversal Verification's losslessness in the paper, as well as a proof of its optimality when the tree degenerates into a chain.

The reviewers highlight our strengths as follows:

- Significant Practical Relevance: "The problem itself is important and practical, as most of the best-performing speculative decoding methods use a tree-based speculation structure."  (Reviwer VunK) / "There is indeed potential waste in the token tree. If these latent tokens can be effectively utilized, there is a chance for some improvement." (Reviwer QRvg)
- Solid Theoretical Foundation: "The paper provides rigorous proofs for lossless guarantee and optimality in specific scenarios." (Reviewer VhHZ) / "There is a theoretical argument demonstrating that Traversal Verification is lossless." (Reviwer QRvg)
- Comprehensive Experimental Validation: "The experimental validation is comprehensive." (Reviwer mJgv) / "solid empirical evaluations are a plus." (Reviwer VunK)
- Valuable contribution: "This work represents sufficient progress in the field and makes a valuable contribution to speculative decoding research." (Reviewer mJgv) / "The proposed method can be easily adopted to any speculative decoding framework." (Reviwer VunK)

The reviewers' questions primarily focused on the following aspects:

- Experimental validation of Traversal Verification's losslessness (Reviewer VhHZ, Reviewer QRvg) - We've provided empirical results on generation quality of Traversal Verification.
- Experiments on more baselines and generalization (Reviewer QRvg) - We've provided additional experiments on EAGLE-vicuna models.
- Technical details and the rigor of some wording in the paper (Reviewer VhHZ) - We've provided explanations to the technical details of our method and will carefully check our wording in our final version to ensure its clarity.
- Stronger mathematical proofs and open questions (Reviewer mJgv, Reviewer VunK) - We've shared our opinions and ideas on these questions.

We are pleased that our explanations and additional experiments sufficiently addressed the reviewers' concerns, and have been unanimously acknowledged by the reviewers.

We will incorporate the relevant discussions and experimental results into the final version of the paper and will carefully check the wording to ensure rigor. The reviewers' valuable feedback has undoubtedly improved the quality of our paper.

Thank you again for your valuable suggestions and recognition of our paper!

---

### Decision · Program_Chairs · 2025-09-17

**Decision:**

Accept (poster)

**Comment:**

This paper introduces Traversal Verification, a novel and interesting leaf-to-root verification method for tree-based speculative decoding. The core idea of utilizing sequence-level probabilities to improve acceptance length, as opposed to the traditional token-level top-down approach, is well-motivated and technically sound. The reviewers unanimously acknowledged the paper's strengths, including the practical importance of the problem (Reviewers Vunk, QRvg), the solid theoretical guarantees of losslessness (Reviewers VhHZ, mjgv), and the comprehensive experimental validation (Reviewer mjgv). Initial concerns were raised by Reviewers VhHZ and QRvg regarding the need for empirical validation of the lossless claim and the inclusion of more baselines to demonstrate generalizability. However, the authors provided a thorough rebuttal with additional experiments, including generation quality scores on MT-Bench and results on the EAGLE-vicuna model, which successfully addressed these concerns and led the reviewers to raise their scores. After the discussion phase, there is a clear consensus that this work represents a valuable contribution to the field of speculative decoding. Therefore, I recommend accepting this paper for a poster presentation.